

# A simplified non-linear chemistry-transport model for analyzing $NO_2$ column observations

Dien Wu[1], Joshua L. Laughner[2], Junjie Liu[2,1], Paul I. Palmer[3,4,2], John C. Lin[5], and Paul O. Wennberg[1,6]

[1]Division of Geological and Planetary Sciences, California Institute of Technology, Pasadena, USA
[2]Jet Propulsion Laboratory, California Institute of Technology, Pasadena, USA
[3]School of GeoSciences, University of Edinburgh, Edinburgh, UK
[4]National Centre for Earth Observation, University of Edinburgh, Edinburgh, UK
[5]Department of Atmospheric Sciences, University of Utah, Salt Lake City, USA
[6]Division of Engineering and Applied Science, California Institute of Technology, Pasadena, USA

**Correspondence:** Dien Wu (dienwu@caltech.edu)

**Abstract.** Satellites monitoring air pollutants (e.g., nitrogen oxides, $NO_x = NO + NO_2$) or greenhouse gases (GHGs) are widely utilized to understand the spatiotemporal variability and evolution of emission characteristics, chemical transformations, and atmospheric transport over anthropogenic "hotspots". Recently, the joint use of space-based long-lived GHGs (e.g., carbon dioxide, $CO_2$) and short-lived pollutants has made it possible to improve our understanding of emission characteristics. Some previous studies, however, lack consideration of the non-linear $NO_x$ chemistry or complex atmospheric transport. Considering the increase in satellite data volume and the demand for emission monitoring at higher spatiotemporal scales, it is crucial to construct a local-scale emission optimization system that can handle both long-lived GHGs and short-lived pollutants in a coupled and effective manner. This need motivates us to develop a Lagrangian chemical transport model that accounts for $NO_x$ chemistry and fine-scale atmospheric transport (STILT-$NO_x$); and investigate how physical and chemical processes, anthropogenic emissions, and background may affect the interpretation of tropospheric $NO_2$ columns (t$NO_2$).

Interpreting emission signals from t$NO_2$ commonly involves either an efficient statistical model or a sophisticated chemical transport model. To balance computational expenses and chemical complexity, we describe a simplified representation of the $NO_x$ chemistry that bypasses an explicit solution of individual chemical reactions while preserving the essential non-linearity that links $NO_x$ emissions to its concentrations. This $NO_x$ chemical parameterization is then incorporated into an existing Lagrangian modeling framework that is widely applied in the GHG community. We further quantify uncertainties associated with the wind field and chemical parameterization and evaluate modeled columns against retrieved columns from the TROPO-spheric Monitoring Instrument (TROPOMI v2.1). Specifically, simulations with alternative model configurations of emissions, meteorology, chemistry, and inter-parcel mixing are carried out over three US power plants and two urban areas across seasons. Using EPA-reported emissions for power plants with non-linear $NO_x$ chemistry improves the model-data alignment in t$NO_2$ (a high bias of $\leq 10\%$ on an annual basis), compared to simulations using either EDGAR or without chemistry (bias approaching 100%). The largest model-data mismatches are associated with substantial biases in wind directions or conditions of slower atmospheric mixing and photochemistry. More importantly, our model development illustrates (1) how $NO_x$ chemistry affects the relationship between $NO_x$ and $CO_2$ in terms of the spatial and seasonal variability and (2) how assimilating t$NO_2$ can





quantify systematic biases in modeled wind directions and emission distribution in prior inventories of $NO_x$ and $CO_2$, which
laid a foundation for a local-scale multi-tracer emission optimization system.

## 1  Introduction

Emissions of air pollutants (APs) and greenhouse gases (GHGs) adversely impact urban ecosystems and environments, human
health, and the climate via the moderation of energy budgets (Myhre et al., 2014; Watts et al., 2021). APs and GHGs are
directly inter-connected considering they are co-emitted from many combustion sources, suggesting that reductions in GHGs
may bring co-benefits in mitigating APs (Cifuentes et al., 2001; West et al., 2013; Lin et al., 2018). Although quantifying
emissions in GHGs and APs and understanding their underlying drivers at all scales are equally important, emission estimates
beyond a county or city become more relevant in addressing policy-relevant topics such as emission mitigation.

Space-based remote sensors offer an objective perspective to monitoring global air quality and GHGs. These new data enable
us to uncover the spatial variability along with the temporal trend and perturbation of anthropogenic emissions. Air quality-
related observations have been among the first to demonstrate the capability of satellite remote sensing to globally diagnose air
quality (Duncan et al., 2016; Laughner and Cohen, 2019; Jin et al., 2020), constrain emissions across time, space, and sectors
(Jiang et al., 2018; Goldberg et al., 2019; Tang et al., 2019; Qu et al., 2022), and evaluate real-world decisions (Lamsal et al.,
2011; Demetillo et al., 2020). Leveraging satellite observations in understanding the spatiotemporal distribution of emissions
within cities is still limited compared to those city-total estimates. Data and analysis uncertainty further present the main
challenge in extracting robust combustion signals from remotely sensed measurements and these uncertainties are amplified in
attempts to resolve dynamic flows and heterogeneous combustion activities within cities (Valin et al., 2013; Goldberg et al.,
2022; Souri et al., 2022).

Making full use of existing and upcoming satellites that retrieve concentrations of APs and GHGs offers an informative way
to target urban emissions from different sources at a policy-relevant scale of a few km. Combining satellite observations of
species with different atmospheric lifetimes has enabled studies to diagnose chemical conditions and meteorological processes
(Jin et al., 2017; Lama et al., 2022), identify urban plumes, and constrain emissions for the tracer of interest (Wunch et al., 2009;
Yang et al., 2023), and obtain observation-based ratios between tracers (Silva and Arellano, 2017; Wu et al., 2022; MacDonald
et al., 2022) to infer structural changes in combustion activities (Reuter et al., 2014; Miyazaki and Bowman, 2023). In light
of the rapid rise in satellite data volume, it is beneficial to have an analysis system that adequately accounts for the important
local-scale processes in interpreting the abundance of GHG and APs in a coupled manner (**Fig. 1**). Analogies to such local-scale
systems in a global context include the AP-focused Tropospheric Chemical Reanalysis (TCR-2, Miyazaki et al., 2020) and the
GHG-focused Carbon Monitoring System-Flux (CMS, Hurtt et al., 2022). Only a few recent multi-tracer modeling systems
aim to bridge $CO_2$ and $NO_2$ column measurements (Reuter et al., 2014; Kaminski et al., 2022; Hakkarainen et al., 2023),
albeit limits in their modeling tools (elaborated in the next paragraph). In addition, as emphasized in Reuter et al. (2014), most
multi-tracer studies rely on emission ratio/conversion ratio from inventories, which can be problematic.



In efforts to interpret $CO_2$ or $NO_x$ emission signatures from satellite observations, most prior studies used either statistical or inversion approaches. The former approach involves the use of Gaussian plume or Exponentially-Modified Gaussian (EMG) models with input from simple wind information to derive emissions of $CO_2$ and $NO_x$ (or lifetime if for $NO_x$) purely from observations in a computationally efficient manner without relying much on prior assumptions of emissions (Nassar et al.,

2022; Beirle et al., 2011). These statistical approaches only provide a plume-integrated emission estimate that can be sensitive to the input wind speed and chemical lifetime. Multiple satellite overpasses need to be aggregated with wind direction aligned for a robust fit in the EMG model to obtain emission and lifetimes. It is challenging to infer and evaluate sub-grid cell variations in emissions. The more sophisticated inverse approach involves the use of a chemical transport model (CTM) that comprehensively accounts for atmospheric transport and chemical transformation and a coupled inversion or data assimilation

system (e.g., Liu et al., 2022; Qu et al., 2022). CTMs are, however, computationally expensive and often involve hundreds of species and their coupling reactions. Most CTMs used in AP-related studies are Eulerian models, which may suffer from complications caused by rigid model grids (Wohltmann and Rex, 2009; Valin et al., 2011). Motivated by these approaches that rely on a constant lifetime or solve for individual chemical reactions, we have built a modeling framework to balance the advantages and imperfections —i.e., to simplify the chemical transformation process that preserves the non-linear relationship

between $NO_x$ emissions and the observed concentration field together with a high-resolution atmospheric transport using a Lagrangian Particle Dispersion Model (LPDM).

LPDMs have been increasingly utilized for emission estimates over the past decades. For instance, the Stochastic Time-Inverted Lagrangian Transport Model (STILT, Lin et al., 2003) building upon HYSPLIT (Stein et al., 2015) has been well adapted to analyze emission signals from all sorts of measurement platforms. STILT was designed to better describe the

movement of air parcels only relevant to an observation site and explicitly provide the source-receptor relationship (i.e., the Jacobian matrix) to facilitate efficient atmospheric inversions for optimizing emissions. Besides, LPDMs themselves possess inherent numerical and computational advantages, such as avoiding artificial smoothing of concentration fields by spurious numerical diffusion in confined model boxes (Wohltmann and Rex, 2009; Lin et al., 2013). More importantly, the Lagrangian transport perspective is intuitively coupled with box models that handle chemical reactions. Noticeable examples include

STOCHEM (Collins et al., 1997), ATLAS (Wohltmann and Rex, 2009), CLaMS v2.0 (Konopka et al., 2019), and HYSPLIT-based variations including HYSPLIT-CheM (Stein et al., 2000), ELMO-2 for ozone (Strong et al., 2010), and STILT-chem (Wen et al., 2012). These Lagrangian chemical models describe the chemical reactions of each species or lumped group with similar functional groups to calculate chemical transformation along trajectories but vary in the complexity of implemented chemistry and parameterization for turbulent mixing and numerical diffusion. Despite these prior modeling efforts, Lagrangian

chemical models are more often adopted to inform the origins of APs but are less commonly used to constrain emissions. Such under-appreciation is in part a result of the heavy computational expenses in solving chemical changes at high frequency via ordinary differential equations (similar to most Eulerian CTMs) and the reliance on external meteorological fields.

To reduce computational costs in dealing with complex chemistry, studies have proposed machine learning techniques or defaulted to a constant-lifetime assumption as a shortcut. Machine learning techniques have been applied to approximate the

chemical mechanisms (Keller and Evans, 2019; Huang and Seinfeld, 2022), predict OH field with observational constraint





(Zhu et al., 2022), and calculate emissions (He et al., 2022). Other studies have assumed a constant first-order lifetime to estimate $NO_x$ emissions and emission ratios between $NO_x$ and $CO_2$ (Lee et al., 2014; Hakkarainen et al., 2023). However, unlike chemically passive species such as $CO_2$, the chemical tendency of $NO_x$ is not independent of atmospheric advection and turbulent mixing because of the chemically-driven non-linearity between the $NO_x$ lifetime and the NO and $NO_2$ concentrations

(Laughner and Cohen, 2019). More specifically, during the day $NO_x$ is lost through two more permanent pathways of (1) $NO_2$ + OH to nitric acid and (2) NO + peroxy radicals ($RO_2$) with a minor branch in producing alkyl nitrates, ANs (**POINT 3 in Fig. 1**). The two pathways compete with one another and either may dominate depending on chemical conditions. Such non-linear dependence of $NO_x$ lifetime or chemical tendency with $NO_x$ concentration must be accounted for to estimate $NO_x$ emissions from atmospheric $NO_2$ concentrations. Such non-linearity will affect the interpretation of tracer-to-tracer emission ratios from

observed enhancement ratios.

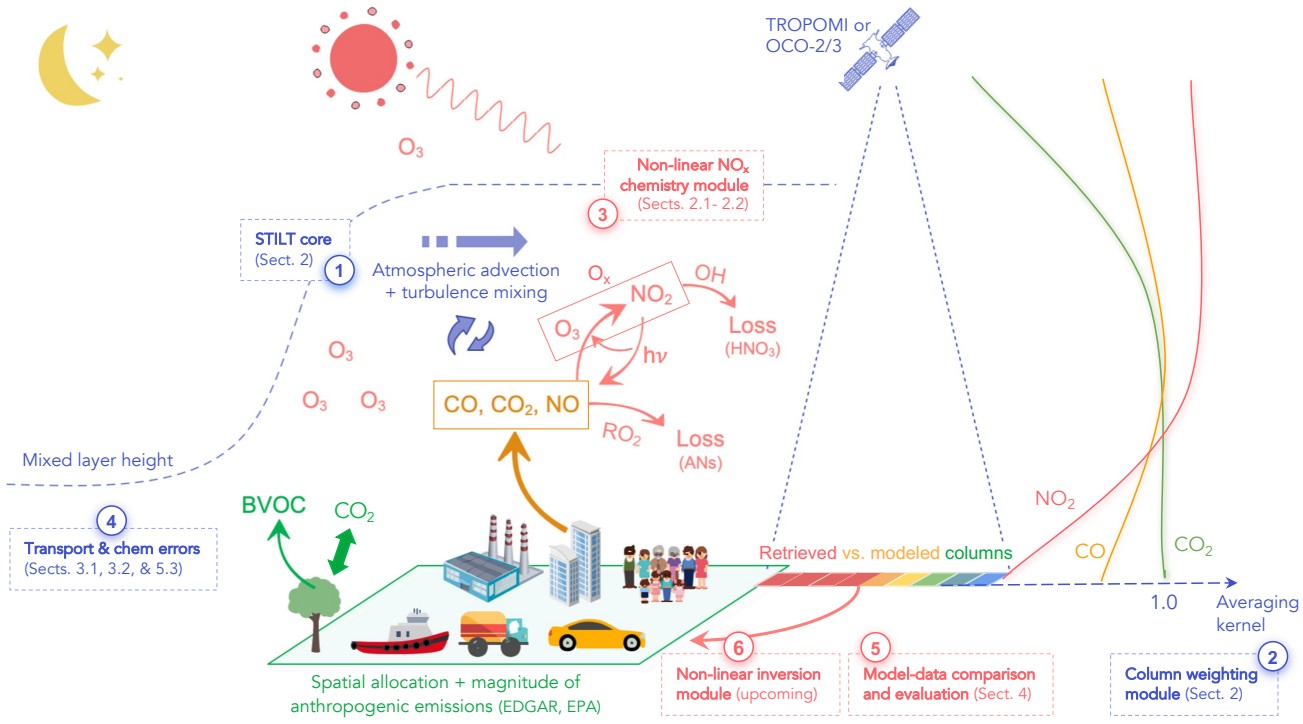

**Figure 1.** A conceptual diagram of our proposed local-scale multi-tracer modeling framework in interpreting column observations. It contains a road map for this study (POINTs 1 through 5). The diagram highlights key biogenic/physical/chemical processes for quantifying $NO_x$, CO, and $CO_2$ around cities based on space-based measurements (pixels from red to blue): atmospheric conditions (wind speed & PBLH for vertical mixing, horizontal mixing/diffusion lengths), chemical conditions (photolysis rate and $NO_x$ regimes, regional versus local oxidant conditions), the spatial distribution of emissions (urban vs. power plant), and sensitivities of the column abundance to individual vertical levels (averaging kernel).

In this study, we present a non-linear modeling framework, STILT-$NO_x$, to simulate tropospheric $NO_2$ columns (t$NO_2$) as retrieved from TROPOMI. As illustrated in **Fig. 1**, the overarching goal of this framework is to facilitate emission optimizations over global anthropogenic hotspots by simulations of the concentrations of key trace gases of $CO_2$, CO, and $NO_x$ at the local



scale. To do so, the current work aims to equip the STILT model with simplified chemistry that avoids explicit calculations of chemical reactions while preserving the non-linearity that ties the $NO_x$ concentrations to its emission (**POINT 3 in Fig. 1**). The proposed STILT-$NO_x$ framework is comprised of four components, which correspond respectively to points 1 to 4 in **Fig. 1** and will be coupled to an upcoming non-linear flux inversion module (**POINT 6**).

1. the HYSPLIT-STILT core that resolves fine-scale atmospheric advection and turbulence; and calculates the sensitivity of concentration anomalies to upwind fluxes ("footprint") (Lin et al., 2003; Fasoli et al., 2018; Loughner et al., 2021); with an additional simplified inter-parcel mixing scheme (**Sect. 2.3**);

2. a column weighting module to simulate atmospheric columns (and uncertainties) that incorporates pressure weighting functions and retrieval-specific averaging kernel profiles (X-STILT, Wu et al., 2018);

3. a simplified chemistry module that describes $NO_x$ chemical tendency (**Sect. 2.1**) and how much $NO_x$ is presented as $NO_2$ ($NO_2$–to–$NO_x$ ratio, **Sect. 2.2**);

4. an error analysis module that quantifies errors and biases in wind fields and chemical parameters (**Sect. 3**) following methods initially proposed in Lin and Gerbig (2005) and Wu et al. (2018), which can be used for future flux inversions.

We illustrate the skill of this framework using comparisons of modeled $tNO_2$ and those diagnosed from TROPOMI over 3 US power plants and 2 cities across seasons (**Sect. 4**). Lastly, we discuss possible future advances in **Sect. 5.3** and demonstrate the benefits of applying this framework, especially on the quantification of $CO_2$ emissions, emission ratios between $NO_x$ and $CO_2$, and "near-field" wind biases in **Sects. 5.1 and 5.2**.

## 2   STILT-$NO_x$ model descriptions

Building upon the HYSPLIT-STILT atmospheric transport core, the STILT-$NO_x$ framework traces the origin of the atmospheric column observed by the satellite and calculates changes of $NO_x$ concentrations due to emissions, inter-parcel mixing, and chemical transformations at the (sub-)minute scale. The STILT-$NO_x$ simulations are conducted in three steps (**Fig. 2**).

First, the backward-trajectory mode records the lat/long/pressure coordinates of air parcels originating from the same atmospheric column sampled by satellites and being driven by the Eulerian meteorological fields (**STEP 1 in Fig. 2**). In this work, we tested two meteorological fields when they are available for each examined region, namely from the Global Forecast System (GFS0p25) and the High-Resolution Rapid Refresh (HRRR) with a respective horizontal grid spacing of $0.25°$ and 3km (Rolph et al., 2017). As most anthropogenic and all soil sources of $NO_x$ are from the surface, air parcels are evenly distributed and released from the surface to 2 km which is slightly above the typical planetary boundary layer (PBL) height (Wu et al., 2018). To evaluate how representative enhancements between 0 and 2 km are compared to the total tropospheric column enhancements (which can include sources from lightning and aviation), we analyzed vertical distributions of $NO_x$ mixing ratios from TCR-2 (Miyazaki et al., 2020). TCR-2 is a global chemical reanalysis that includes full physical and chemical processes for various species and assimilates multiple satellite products of $NO_2$, ozone, CO, and $SO_2$. As a result, monthly mean $NO_x$ concentrations over the $2° \times 2°$ area around the top 1000 cities is quite insignificant for pressure $\leq 700$ hPa compared to huge signals within



the PBL (**Supplement Fig. S1**). Although 0 to 2 km columns include most anthropogenic enhancements over urban areas, we subtracted a local $NO_2$ background from the total tropospheric columns to minimize the non-anthropogenic influences with a plume detection algorithm following Kuhlmann et al. (2019). The model-data comparisons with background subtracted are discussed in **Sect. 4.1**.

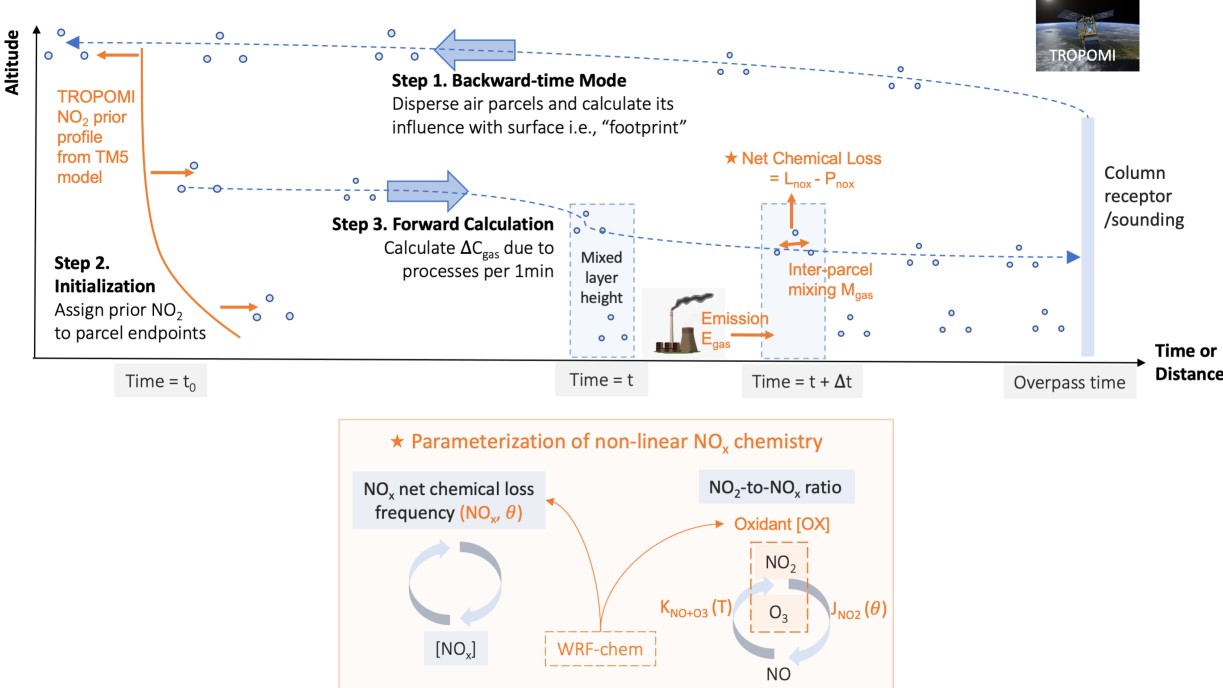

**Figure 2.** A schematic of STILT-$NO_x$ for simulating concentrations in three steps. **STEP 1** — routine backward-time calculation: record locations of air parcels at each timestamp ($\Delta t$) of 1 min or less and their influence from potential fluxes ("footprint"). **STEP 2** — initial condition: the trajectory endpoint at time = $t_0$ is given a concentration from 4D fields (e.g., TM5 in the case of $NO_x$). **STEP 3** — forward-time concentration calculation: updates change in concentrations due to emissions, net chemical losses, and inter-parcel mixing along each trajectory at a timescale of $\leq 1$ min. To clarify, **STEP 3** made use of trajectories originating from a column receptor stretching from the surface to 2 km generated from **STEP 1**.

After being released from a given TROPOMI sounding at the overpass time ($\sim$1 pm local time for nadir soundings), air parcels are dispersed backward in time for 12 hours (time at $t_0$ in **Fig. 2**). **STEP 1** also provides the STILT "footprint" [ppm/($\mu$mol m$^{-2}$ s$^{-1}$)] per air parcel per timestamp (Lin et al., 2003). STILT footprint of a given air parcel is proportional to the time this parcel spends in a small area (of $\sim$100 meters) and describes how the downwind concentration may be altered if this air parcel is influenced by emissions. A much more complete description of STILT can be found in Lin et al. (2003);



Fasoli et al. (2018). The footprint concept, by definition, relies on atmospheric transport and only accounts for concentration changes due to emissions, but not chemical transformations.

Next, $NO_x$ concentrations at the endpoints of the model trajectory are extracted from the Tracer Model version 5, Massively Parallel version (TM5-MP) to serve as the initial conditions (**STEP 2 in Fig. 2**). TM5-MP is an auxiliary dataset whose $NO_2$ vertical profiles serve as the prior knowledge facilitating the stratosphere-troposphere separation in L2 $NO_2$ retrieval (Van Geffen et al., 2022). Here, we simply assume that most $NO_x$ is presented as $NO_2$ at nighttime, despite the apparent caveat in neglecting $NO_3$ chemistry and heterogenous reactions involving $N_2O_5$.

Once $NO_x$ is initialized at the time $t_0$ for the endpoint of every trajectory, we proceed with (**STEP 3 in Fig. 2**) to estimate changes in concentrations due to emissions, chemical transformation, and inter-particle mixing. Mathematically, the concentration per air parcel per timestamp ($C_{p,t}$) relies on that from the last timestamp following **Eq. 1**:

$$C_{p,t} = C_{p,t-\Delta t} + \Delta C_{emis,p,t}(E, F_{p,t}) + \Delta C_{chem,p,t}(C_{p,t}, \theta_{p,t}) + \Delta C_{mix,p,t}(C_{p,t}, \overline{C_{p_{ngb},t}}) \qquad (1)$$

where the time interval for updating concentrations, $\Delta t$, is defaulted to 1 min or reduced to sub-minute when $C_t$ becomes nonphysically negative to ensure numerical stability. Concentration gains from emissions, $\Delta C_{emis}$, result from multiplying STILT parcel-specific footprints ($F_{p,t}$) with prior emissions (E) from EDGARv6.1 (Crippa et al., 2022) and EPA (United States Environmental Protection Agency, 2022) for power plant cases in this study. We neglect soil $NO_x$ emissions given the relatively small contributions in cities. Unlike sophisticated CTMs which resolve chemical reactions of an individual or lumped groups of species, concentration anomalies due to chemical reactions, $\Delta C_{chem}$, are solved in an explicit first-order fashion involving a "net chemical tendency" with a unit of ppb $hr^{-1}$. Such a chemical tendency ($R_{NO_x}$ in **Eq. 2b**) is parameterized offline as functions of $NO_x$ concentrations and solar zenith angles, $\theta$, which is explained in **Sect. 2.1**. The final term, $\Delta C_{mix,p,t}$, denotes the concentration exchange between a given air parcel and its volumetric neighborhood ($p_{ngb}$), which is explained in **Sect. 2.3**.

Following these steps, we obtain modeled $NO_x$ concentrations in ppb for every column receptor between the surface and 2 km. To compare against TROPOMI tropospheric $NO_2$ columns, we need to account for the fraction of $NO_x$ that is present as $NO_2$ (**Sect. 2.2**) and properly weight modeled $NO_2$ from different altitudes according to pressure weighting function and averaging kernel profiles following Wu et al. (2018). Such an approach in applying averaging kernel (**Fig. 1**) to modeled profiles is equivalent to a more commonly used approach, which re-calculated retrieved tNO$_2$ as "seen" from the CTM by re-calculating air mass fraction based on modeled $NO_x$ profiles as investigated in Goldberg et al. (2022). In addition, we evaluate the modeled meteorology and chemistry using a separate set of STILT-$NO_x$ simulations with "true" $NO_x$ emissions from EPA for three US power plants (**Sect. 4.1**).

## 2.1  $NO_x$ net chemical tendency, $R_{NO_x}$, and uncertainty

Inspired by the theoretical non-linear curves of $NO_x$ lifetimes as functions of $NO_2$ vertical column density and volatile organic compound reactivity ($VOC_R$) based on a box model in Laughner and Cohen (2019), we extract similar non-linear parameterizations using the Weather Research and Forecasting model coupled with Chemistry (WRF-Chem v4.0.2, Grell et al., 2005). Considering urban areas being the main focus of our study, we carried out WRF-Chem simulations over three mid-latitude





cities and focused on outputs over a $2° \times 2°$ region around the city center to spotlight chemical regimes in urban environments. The three cities are Los Angeles in the US, Shanghai in China, and Madrid in Spain, which varied in $NO_x$ and VOC emissions

and climatology. **Appendix A** describes our specific WRF-Chem settings used to generate look-up tables of $NO_x$ chemical loss tendencies, which we will refer to as "$NO_x$ curves" (**Fig. 3**). Of the WRF-Chem settings, the chosen chemical mechanism (RADM2, Stockwell et al., 1990) is the most relevant to the accuracy of these $NO_x$ curves. Despite uncertainties in these WRF-Chem simulations, what matters the most for reproducing the $NO_x$ tendency is how $NO_x$ varies with for example solar zenith angle and ozone, rather than the exact accuracy of $NO_x$ concentrations themselves (from WRF-Chem). Thus, non-chemical

components (prior emissions, boundary conditions, and physical processes) in this specific WRF-Chem configuration do not necessarily need to be "perfect" or optimized against observations. We clarify that WRF-Chem simulations had been performed to facilitate the parameterization of $NO_x$ tendency within STILT-$NO_x$ but are not required when running STILT-$NO_x$.

By leveraging WRF-Chem's chemical diagnostic feature, we derive the net chemical tendency of $NO_x$ within each hour $[R_{NO_x}, ppb\ hr^{-1}]$ per model grid (x, y) based on the output cumulative chemical changes in NO and $NO_2$ (i.e., "chem_no2"

and "chem_no" in WRF-Chem registry) following **Eqs. 2**:

$$\sum_{h_0}^{h} \Delta C_{NO_x}(x,y) = \sum_{h_0}^{h} \Delta C_{NO}(x,y) + \sum_{h_0}^{h} \Delta C_{NO_2}(x,y) \tag{2a}$$

$$R_{NO_x}(x,y,h) = P_{NO_x}(x,y,h) - L_{NO_x}(x,y,h) = \frac{\sum_{h_0}^{h} \Delta C_{NO_x}(x,y) - \sum_{h_0}^{h-1} \Delta C_{NO_x}(x,y)}{1\ hr} \tag{2b}$$

where model hour h denotes the beginning time of the hour interval of the hourly WRF-Chem outputs. $\sum_{h_0}^{h} \Delta C_{NO_x}$ describes the cumulative net changes to $NO_x$ concentration given chemical reactions from the initial model hour $h_0$.

WRF-Chem pixel-specific hourly $NO_x$ rate changes, $R_{NO_x}$, are then grouped by both SZA ($\theta$) bins with a spacing of $2°$ and $C_{NO_x}$ bins with equal spacing in log10 scale (**Fig. 3a**). $\theta$ is chosen given the close relation to solar radiation under clear-sky conditions and controls the photolysis frequency of ozone and OH production (Rohrer and Berresheim, 2006) when ozone and water vapor abundance remain unchanged. Because the intention in using STILT-$NO_x$ is to inform the relationship between emission sources and satellite $NO_2$ columns, which are almost always filtered to remove cloudy scenes (i.e., quality assurance

of $\geq 0.7$), the choice of $\theta$ without considering cloud coverage is reasonable. Specifically, these net chemical changes explicitly contain all $NO_x$-relevant reactions within the WRF-Chem/RADM2 scheme, such as the recycling of $NO_x$ from oxidized odd-nitrogen species like peroxyacetyl nitrate.

As a net result, $R_{NO_x}$ is mostly negative during the day, meaning $NO_x$ is removed from the system. $R_{NO_x}$ is large with small spread at low $\theta$ of $\leq 20°$ and gradually decreases during the day. $R_{NO_x}$ becomes positive as approaches nighttime hours

(**Supplement Fig. S2**) and its variability peaks during sunset when $\theta \in [80°, 100°]$ with a fractional uncertainty of over 100% (blue error bars in **Figs. 3a**) considering the transition to nighttime chemistry. When focusing on the daytime portion with $\theta < 70°$ and $C_{NO_x} \geq 1$ ppb, the spread in $R_{NO_x}$ among WRF-Chem urban pixels ranges from 12.2% to 67.9% according to varied $\theta$ and $C_{NO_x}$ (red to yellow error bars in **Fig. 3a**) with an average uncertainty of 41.2%. When focusing on the nighttime portion with $\theta \geq 70°$ and $C_{NO_x} \geq 1$ ppb, the spread in $R_{NO_x}$ spans from 27.9% to over 100% with an average uncertainty of 96.3%

largely skewed by the high uncertainty around the dusk hours. Lastly, the average daytime uncertainty in the $NO_x$ tendency at





medium to high $NO_x$ concentrations (i.e., 41.2%) will be propagated into chemical uncertainties in $tNO_2$ for cases of power plants and urban areas, which is further described in **Sect. 3**.

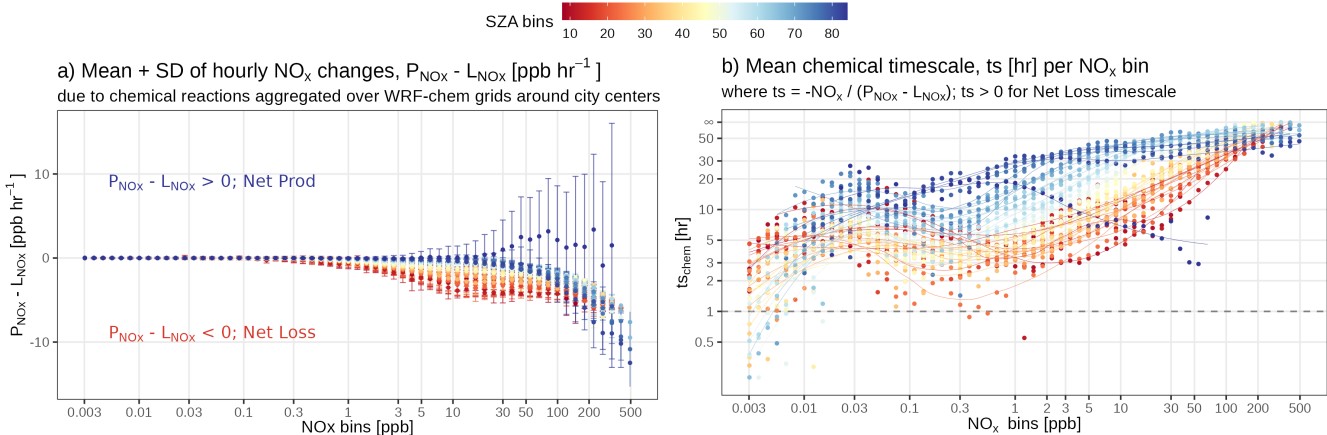

**Figure 3.** A diagram of $NO_x$ net chemical loss tendency [$R_{NO_x}$, ppb hr$^{-1}$] as functions of $NO_x$ concentration ($C_{NO_x}$) and solar zenith angle ($\theta$). The net loss timescale was first calculated from each 12 km WRF-Chem grid cell and aggregated into multiple bins of $NO_x$ concentration [ppb]. The $NO_x$ bins are equally divided in the logarithmic space. The solid dots and error bars denote the average and standard deviation of $R_{NO_x}$ within each combined $\theta$ and $C_{NO_x}$ bin. For the net loss timescale, only positive values are displayed given the logarithmic scale of the y-axis in panel b.

Given the further fluctuation in $R_{NO_x}$ with $C_{NO_x}$, we define a "net loss timescale" [hr] as $ts_{NO_x} = -C_{NO_x}/R_{NO_x}$ and distinguish it from the conventional chemical lifetime that only accounts for chemical losses. For reference, a positive (or neg-

ative) timescale corresponds to a net loss (or production) of $NO_x$ (**Fig. 3b**). The contribution from $NO_x$ production is minor during noon hours. The non-linear dependence of $ts_{NO_x}$ with $C_{NO_x}$ is largely driven by several $NO_x$ loss pathways: predominately by the loss processes of $NO_2 + OH$ and the formation of alkyl nitrates during the daytime and by the $NO_3$ chemistry and heterogeneous chemistry at nighttime (**Supplement Fig. S2**). Here, we do not differentiate $NO_x$ curves by $VOC_R$ despite its critical role in determining the turning point when $NO_x$ is mainly lost to either nitric acid or alkyl nitrates (Laughner and

Cohen, 2019). We instead perform a sensitivity study of the impact on $NO_x$ curves for three $VOC_R$ intervals in **Sect. 5.3**. Note that these $NO_x$ curves should be considered as a first-order approximation and can certainly be improved upon to evaluate more complex parameterization (**Sect. 5.3**). When it comes to calculating chemical changes within STILT-$NO_x$ per air parcel per timestamp (i.e., $\Delta C_{chem,p,t}$ in **Eq. 1**), such a loss timescale is looked up according to parcel-specific $\theta$ and $C_{NO_x}$ to enable the non-linearity core (**Fig. 2** bottom).

**2.2 $NO_2$–to–$NO_x$ ratio**

As only the vertical column density of $NO_2$ is retrieved, the fraction of $NO_x$ present as $NO_2$ as TROPOMI passed over is an important component of our analysis. Prior studies estimated such ratios using a constant value (e.g., of 0.75) at noon hours across seasons with a 10% uncertainties (Beirle et al., 2011, 2019; Goldberg et al., 2022), monthly mean climatology of ozone from reanalysis (Beirle et al., 2021), and CTMs. $NO_x$ is primarily emitted as NO but converted to $NO_2$ via the reaction with



ozone. During the daytime, $NO_2$ is photolyzed back to NO with a photolysis frequency, $J_{NO_2}$. Thus, $NO_2$–to–$NO_x$ ratio scales with the ratio of ozone and $J_{NO_2}$ (**Eq. 3a**).

Considering the close coupling between $NO_2$ and $O_3$, their sum $O_x$ in **Eq. 3b**, is a key indicator for atmospheric oxidant capability in understanding urban air chemistry (Clapp and Jenkin, 2001; Fujita et al., 2016) and informing chemical dynamics (e.g., during COVID, Parker et al., 2020; Lee et al., 2020). $O_x$ levels within PBL can be regarded as a $NO_x$–independent
component related to regional ozone inflow plus a $NO_x$–dependent component that non-linearly varies with local $NO_x$ and $VOC_R$ conditions (Clapp and Jenkin, 2001; Jenkin, 2004). The complexity in the local $O_x$-$NO_x$ non-linearity is caused by key reactions behind $NO_x$ curves, which are elucidated in **Sect. 5.3**. For simplification, we prescribed the normal $O_x$ level as 50 ppb and calculate the $NO_2$–to–$NO_x$ ratio via **Eqs. 3** assuming steady-state:

$$J_{NO_2}(\theta)[NO_2] = k_{NO+O_3}(T_A, P)[O_3][NO] \tag{3a}$$

$$[O_x] = [O_3] + [NO_2] \tag{3b}$$

where $J_{NO_2}$ relies on $\theta$ for daytime and the reaction rate coefficient of NO with $O_3$ ($k_{NO+O_3}$) is a function of air temperature $T_A$ and pressure P (**Fig. 2** bottom). The inclusion of $O_x$ in the calculation of $NO_2$–to–$NO_x$ ratio is to avoid non-physical infinite conversion of NO to $NO_2$ at high-emitting sources following the titration of ambient ozone. Sensitive tests were performed to reveal how biases in prescribed $O_x$ level may modify the modeled $tNO_2$ (**Sect. 3**). Typical $NO_2$–to–$NO_x$ ratios over examined
mid-latitude targets across seasons are summarized in **Sect. 5.2**. In the future, satellite observations of tropospheric ozone could be used to add the additional complexity of variable $O_x$.

## 2.3   Inter-parcel mixing

Eulerian chemical models usually suffer from too strong mixing or numerical diffusion within their model grid; while Lagrangian models (equivalent to possess extremely high "spatial resolution") may lack any mixing between air parcels that are
normally assumed to be independent of one another (Lin et al., 2013; Brunner, 2012). Such lack of mixing has negligible impact on passive tracers as mixing alters only the spatial distribution of concentration among air parcels but not the resultant concentration averaged across parcels at the receptor. However, non-linear processes alter both the spatial distribution of parcel-specific concentrations together with the average resultant concentration. As a result, the calculation of the total $NO_x$ tendency will be sensitive to how inter-parcel mixing is parameterized. Common ways to realize turbulence mixing are through (1)
stochastic processes followed by the exchanging/averaging properties of air parcels found within a certain mixing length (e.g., STILT-chem, Wen et al., 2012), (2) implemented deformation- and instability- driven schemes that rely on atmospheric stability and wind shear/stress characteristics (e.g., CLaMS, McKenna et al., 2002; Konopka et al., 2019), and (3) diffusion approaches that require the vertical gradient of concentrations (e.g., CiTTyCAT and ELMO-2, Pugh et al., 2012; Strong et al., 2010).

Here we follow the STILT-chem approach to enable a process of exchanging concentrations per timestamp among the air
parcels in close proximity to each other ($\Delta C_{mix}$ term back in **Eq. 1**), which smooth the concentration gradient among those air parcels. Specifically, at the timestamp of t, the concentration for a given air parcel p is updated based on the concentration gradient between p and its neighborhood according to a mixing timescale ($\tau_{mix}$) within a grid volume with a mixing length



scale of a horizontal area and the mixed layer height for the height as follows:

$$C'_{p,t} = C_{p,t} \exp\left(-\frac{\Delta t}{\tau_{mix}}\right) + \overline{C}_{P_{ngb},t} \left[1 - \exp\left(-\frac{\Delta t}{\tau_{mix}}\right)\right] \tag{4}$$

where $\exp\left(-\frac{\Delta t}{\tau_{mix}}\right)$ implies the degree of mixing and $\overline{C}(t)$ represents the average concentration among air parcels within the mixing volume. The update of $C'_{p,t}$ from $C_{p,t}$ responds to $\Delta C_{mix}$ in **Eq. 1**. A relatively fast mixing time scale of 3 hours and a horizontal mixing length of 1 km is used for testing the mixing impact on modeling $tNO_2$. We neglect the slower mixing in the free troposphere and tested alternative length- and time-scale values for ML parcel-mixing (**Sect. 5.3**).

## 3    Model uncertainty in $tNO_2$ due to wind and chemistry

As atmospheric transport and chemical transformation are the two main components in any CTMs, we assess how uncertainties tied to the modeled wind field, $NO_x$ loss timescale, and $NO_2$–to–$NO_x$ ratio may contribute to uncertainties in $tNO_2$ in ppb.

$$\sigma^2_{sim} = \sigma^2_{trans} + \sigma^2_{ts} + \sigma^2_{nn}. \tag{5}$$

Here we briefly describe how various $tNO_2$ uncertainties were approximated based on our understanding of errors in respective model parameters/inputs (i.e., wind error, $NO_x$ chemical tendency, or $O_x$ levels). To approximate $tNO_2$ uncertainties
due to transport errors, we followed previous approaches to first assess the GFS- and HRRR- modeled wind profiles against radiosonde, calculate respective error statistics including wind error, correlation time and length scales, and lastly propagate wind error statistics into errors in column concentrations. Mathematically, $\sigma^2_{trans}$ in **Eq. 5** is derived from the difference in the variance of STILT-$NO_x$ air parcel-specific $NO_2$ concentrations between the original simulation and a second simulation with wind error (Lin and Gerbig, 2005; Wu et al., 2018). The derivations of modeled wind errors and contributions to $tNO_2$ er-
rors are elaborated in **Appendix B**. To evaluate the impact due to errors associated with chemical parameters, we perturbed the $NO_x$ curves or the $O_x$ level according to 20 perturbing factors. Perturbed curves/parameters are used to generate 20 new sets of $tNO_2$ fields, of which their respective standard deviation among perturbations serves as the chemical uncertainty [ppb] due to $NO_x$ net loss timescale and $NO_2$–to–$NO_x$ ratio ($\sigma_{ts}$, $\sigma_{nn}$ in **Eq. 5**). These 20 perturbing factors were randomly selected from a normal distribution $N(\mu = 1, \sigma_{param})$. Here we tested out $\sigma_{param}$ of 40% for $NO_x$ loss timescales according to uncertainties
in the chemical tendency (**Fig. 3**) and a $\sigma_{param}$ of 40% for $O_x$ level (**Eqs. 3**).

Due to heavy computational expenses in conducting such wind and chemical perturbation analyses for all overpasses and locations, we only ran error analyses for a total of six overpasses over a power plant and a city. To cover seasonal changes in $NO_2$ signals and their uncertainties, overpasses in varied seasons are examined for the New Madrid power plant on Feb 8, June 15, and Dec 8, 2020, and Phoenix on Feb 7, May 27, and Dec 23, 2020. Two winter cases with relatively large signals are shown
in **Fig. 4a**. Considering the non-linearity between chemical tendency and $NO_x$ concentration, sounding-specific uncertainties for all six cases are presented against modeled $tNO_2$ in **Fig. 4b**. When conducting those perturbations, other model parameters like meteorological field and emissions remain unchanged.

As a result, the average percent error in u-/v- wind speed in the PBL is roughly 22% for the New Madrid case (**Supplement Fig. S3a**), which contributes to 50% uncertainty in $tNO_2$ at the sounding level (3rd column in **Fig. 4a**). Higher transport errors



may more frequently occur if an intensive point source is in the area or over pixels on the border of the $NO_2$ plumes with moderate signals of about 0.2 to 0.5 ppb (e.g., dots in **Fig. 4b**). This is because small deviation in modeled wind vectors causes air parcels to either "hit" or "miss" the intensive source. The transport uncertainty appears to first correlate positively with the signals and then decreases when signals are sufficiently high, e.g., > 0.7 ppb. Such a decline may be associated with hyper-near-field soundings, where deviation in wind fields may not alter modeled signals as modeled air parcels will always

experience large influence from the emission source (dotted-dashed lines in **Fig. 4b**). Compared to power plants, cities may be associated with a more homogeneous transport uncertainty if emissions are more homogeneous and better mixed in the PBL.

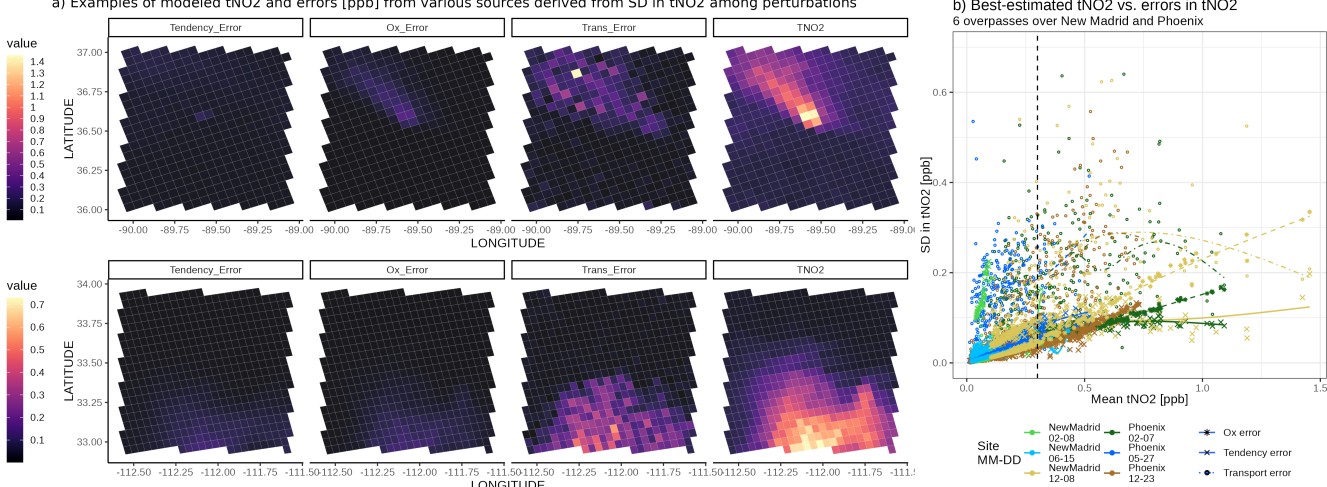

**Figure 4.** (a) Demonstrations of best-estimated $tNO_2$ ("TNO2") and their uncertainties [ppb] due to random u-/v- wind errors ("Trans_Error"), $NO_x$ chemical tendency ("Tendency_Error"), and $O_x$ levels ("Ox_Error") on Dec 8, 2020, over the New Madrid power plant and Dec 23, 2020, over Phoenix. (b) Scaling between uncertainties and mean $tNO_2$ signals over six overpasses for the two targets with smooth splines fitted (crosses with solid lines for tendency errors, stars with dashed lines for Ox errors; and circles with dotted-dashed lines for transport errors). Colors differentiate the sites and TROPOMI overpass times.

     Given a roughly 40% uncertainty in $O_x$ levels or $NO_x$ chemical tendency, chemical uncertainties in ppb remain small when modeled signals are compared (**Fig. 4a**). Uncertainties from chemical tendency first increase with $tNO_2$ signals and gradually plateau for $tNO_2$ beyond 0.7 ppb, likely because $NO_x$ is lost slowly when the $NO_x$ concentration stays high and further

perturbations in chemical tendency are less impactful. In contrast, uncertainty from $O_x$ levels appears to consistently scale against the signals, i.e., more apparent for soundings adjacent to the power plant with reasons explained as follows: When the certain perturbed $O_x$ level approaches zero, the amount of NO that can be oxidized as $NO_2$ becomes minimal ($2^{nd}$ column in **Fig. 4a**). This case mimics the scenario where $O_3$ can be titrated in proximity to an intense release of NO before the ozone-depleted plume air is mixed with the ambient ozone-rich air. Nevertheless, considering the entire sample, the percent errors

due to chemical parameters remains relatively low (13% to 18% for six cases in **Fig. 4b**).

     Whether chemical or meteorological errors dominate the total model errors fundamentally depends on tropospheric $NO_2$ signals which further rely on factors like atmospheric stability with wind errors, chemical tendency, and emission distribution.



Such dependence leads to spatial gradient and seasonal variations in estimated errors as seen from the above examples. In brief, our limited perturbation experiments suggest that transport uncertainties dominate the total modeled uncertainties, except for a few hyper-near-field soundings where chemical uncertainties become more substantial.

For future emission optimizations, uncertainties in the emissions, retrieval, and background should also be included. Despite the significant advance in the TROPOMI $NO_2$ retrieval version of v2 compared to v1 (Van Geffen et al., 2022), v2 retrieval is associated with a fractional uncertainty (normalized over retrieved $tNO_2$) of $\sim$30 to 50% for most soundings within the plume. Uncertainties in $NO_x$ emissions between inventories can serve as the prior uncertainty, which is substantial at the pixel level (**Supplement Figs. S4, S5**). Besides the regional wind assessment, a novel plume rotation algorithm based on model-data $NO_2$ plumes is proposed in **Sect. 5.2** to quantify near-field wind biases.

## 4 Model-data evaluations and comparisons

The tropospheric $NO_2$ mixing ratio at a given sounding location is influenced by the regional inflow, atmospheric advection and turbulence mixing, underlying emission characteristics, and chemical changes en route to the sounding (**Fig. 2**). Modeled tropospheric $NO_2$ mixing ratios using a variety of model configurations are compared against retrieved values from TROPOMI. Such model-data comparisons help evaluate the overall model performance and the roles of individual physical and chemical processes with a naming convention of <MET>_<EMISS>_<GAS>_<PROC> explained as follows:

∗ <MET> represent meteorological fields of either $0.25°$ GFS or 3km HRRR that are used to drive STILT air parcels.

∗ <EMISS> represent two prior $NO_x$ emission inventories. EDGARv6.1 with monthly mean emissions and the latest year available of 2018 is the primary one for simulating all cases. Hourly mean emissions from EPA reports are only used to evaluate modeled chemistry and meteorology for several US power plants (**Sect. 4.1**).

∗ <GAS> represent the simulated species with a default string of "TNO2" without subtracting a localized $tNO_2$ background. A separate comparison with background subtracted is shown in multi-track comparisons (**Fig. 6**, **Sect. 4.1**).

∗ <PROC> denotes the physical and chemical processes considered per run. Two main configurations include (1) "DEF" runs with both inter-parcel mixing and chemical parameterization included, and (2) "NOCHEM" runs with mixing but without considering the $NO_x$ chemical tendency. The "NOCHEM" runs do account for the $NO_2$–to–$NO_x$ conversion but as a constant ratio of 0.74 according to EMG-based studies.

Only model-data comparisons using TROPOMI v2 are shown. As satellite averaging kernel and observed $tNO_2$ differ substantially between v1 and v2, modeled concentrations are weighted by the version-specific AKs to yield apple-to-apple comparisons. Changes in AKs, retrieved and modeled values between versions are summarized in **Supplement Fig. S6**.

### 4.1 Model validation: US power plants

The New Madrid power plant along the Mississippi River is a 1,300-megawatt coal-fired power station (GEM, 2021), which ranks first in 2020 among US power plants regarding $NO_x$ emissions provided by EPA. Thomas Hill and Martin Lake power



plants ranked second and third in 2020, respectively. We also report results for an overpass over the Intermountain power plant
in Utah where the surrounding complex terrain is difficult to be modeled properly. Let us start with two examples to illustrate
plumes modeled by different model configurations (**Sect. 4.1.1**) and then present model-data comparisons over dozens of
overpasses of the three power plants (**Sect. 4.1.2**).

### 4.1.1 Single-track demonstration

The correction in $NO_x$ emissions greatly improves the model-data alignment. For example, EDGAR-based simulations sub-
stantially underestimate or overestimate the tropospheric columns (**Fig. 5a1, 5c1**), as EDGAR emissions are almost 1/3 or twice
of the reported hourly EPA emissions for New Madrid or Intermountain power plant, respectively (**Supplement Fig. S7**). EPA-
based simulations align better with retrieved values from TROPOMI v2 despite deviations over the far-field region (**Fig. 5a2
vs. 5a6**). Such improvements in model-data alignment are also inferred from the linear regression slopes reported in **Fig. 5bd**.
Not accounting for $NO_x$ chemistry or lifetime elevates $NO_2$ concentrations both within the plume and over the background
even if EPA emissions are assumed to be "correct" (**Fig. 5a3, 5a5**). The inter-parcel mixing with a 3-hour mixing timescale
redistributes $NO_x$ concentrations among adjacent air parcels but leads to a minimal impact of $\leq 5\%$ of the modeled $tNO_2$ at
individual column receptors (thereby not shown).

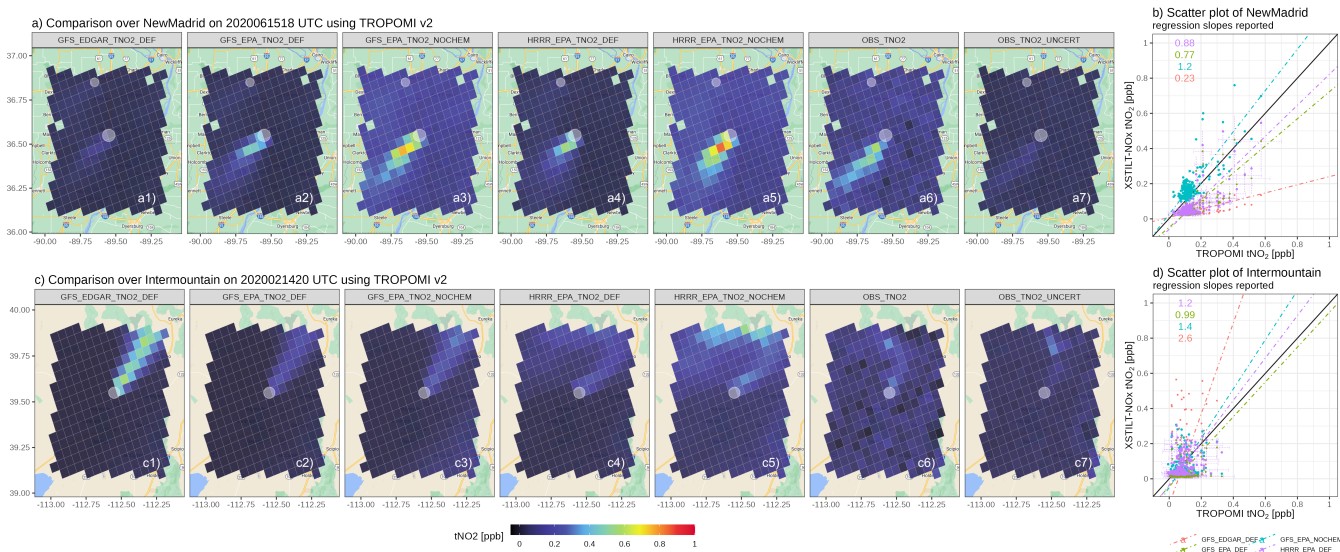

**Figure 5.** Maps (a) and scatter plot (b) of modeled plumes based on several model configurations (first five columns) versus retrieved plumes + uncertainties
from TROPOMI v2.3 (last two columns) for the New Madrid power plant on June 15, 2020, and Intermountain, Utah power plant on Feb 14, 2020. Varied
model configurations are labeled on the top of each panel, following the naming convention of "<MET>_<Emiss>_<GAS>_<PROC>" explained in the
list of **Sect. 4**. In particular, "_DEF" and the "_NOCHEM" denote the modeled columns using the default (with mixing and chemistry) and non-chemistry
configurations. Grid cells with intensive NO emissions from EDGARv6 are labeled as white circles with sizes denoting the relative emission magnitude. The
type II linear regression slope is fitted for each configuration (dotted-dashed line) and modeled and retrieval uncertainties are added (dashed error bars).



The choice of meteorological fields with different spatial resolutions insignificantly affects the modeled signals except for cases surrounded by complex topography and flows. For example, the HRRR-based plumes resemble the GFS-based plumes for the New Madrid power plant (**Fig. 5a2 vs. 5a4**), which is also revealed by their similar wind error statistics (**Supplement Fig. S3a**). However, complex terrain and stable PBL during wintertime complicate and usually worsen the model performance as a result of increased meteorological errors. On Feb 14, 2020, EPA-based plumes over the Intermountain power plant in Utah using two meteorological fields differ substantially from each other and they both deviate from the observed plume regarding the plume shape. GFS delineates the mean wind direction within its coarser $0.25°$ grid box while 3km HRRR offers more spatial variability in wind directions (**Fig. 5c2 vs. 5c4**). Yet, precisely capturing the curvature in the wind vector is extremely challenging even using 3km meteorological fields (**Fig. 5c6**) and more difficult using Gaussian plume approaches that rely on only one effective wind vector. Such model-data mismatch in plume shapes can further help quantify the wind biases, which are discussed in **Sect. 5.2.1**.

Besides modeling challenges, retrieval uncertainty cannot be neglected, as it ranges from 22% to 31% of the retrieved signal for the New Madrid case (**Fig. 5a7**) and up to 100% for the Intermountain case (**Fig. 5c7**) at the sounding level. When using retrieved data and averaging kernel from TROPOMI v1, the regression slope becomes 1.18 and 1.25 (**Supplement Fig. S8**), indicating that modeled plumes using both meteorological fields are larger than observed plumes. While using TROPOMI v2, the respective slopes are 0.88 and 1.2 (**Fig. 5bd**). This again emphasizes the substantial uncertainty in retrieved signals, large enough to even alter the conclusion of whether emissions are underestimated or overestimated for a single overpass; and the need for analyzing multiple overpasses for evaluations (**Sect. 4.1.2**).

### 4.1.2 Multi-track evaluation

To provide a broad impression of the model performance, we expand the model-data comparisons to a total of 50 TROPOMI overpasses across all seasons in 2020 including 34 overpasses for the New Madrid power plant and 9 and 7 summertime overpasses for the Thomas Hill and Martin Lake power plants, respectively. These overpasses are selected based on their relatively intense signals compared to the surrounding. Model-data comparisons for all overpasses are shown on maps in **Supplement Figs. S9 –S12** with linear regression slopes reported and summarized in **Fig. 6** and **Table S1**.

Cases with slopes deviating significantly from 1 are usually associated with substantial near-field wind directional biases. For instance, modeled wind vectors on March 11, April 28, and Sept 9, 2020, have directional biases of > 30 degrees (**Supplement Figs. S9b, S10b**), which explain the respective abnormal linear regression slopes of -1.75, 0.49, and 3.2 (**Fig. 6**). EDGAR-based simulations are biased too high or too low by a factor of two or more compared to observed values from TROPOMI v2.3 (green dots in **Fig. 6**), driven by biases in EDGAR emissions (**Supplement Fig. S7**). The "NOCHEM" simulations without the account of $NO_x$ losses overestimate $tNO_2$ by a factor of two across all seasons and three power plants, regardless of the meteorological or emission fields adopted (empty circles in **Fig. 6**). "Upgrading" meteorological fields to a higher resolution seems to contribute less to the improvement of model-data agreements than "correcting" emissions or chemistry. In the end, modeled values with $NO_x$ chemistry and correct EPA emissions using either GFS or HRRR yield the best agreement with retrieved values from TROPOMIv2 (orange dots and lines in **Fig. 6**). Aggregating results of all overpasses, simulations using





the "best" knowledge of emissions, the simplified chemistry, two different meteorological fields, and inter-parcel mixing are slightly high biased (regression slope up to 1.2, **Table S1**). The RMSE between modeled and observed $tNO_2$ values ranges from 0.11 to 0.15 ppb, which is comparable to the random uncertainty in the $NO_2$ retrieval of 0.09 ppb.

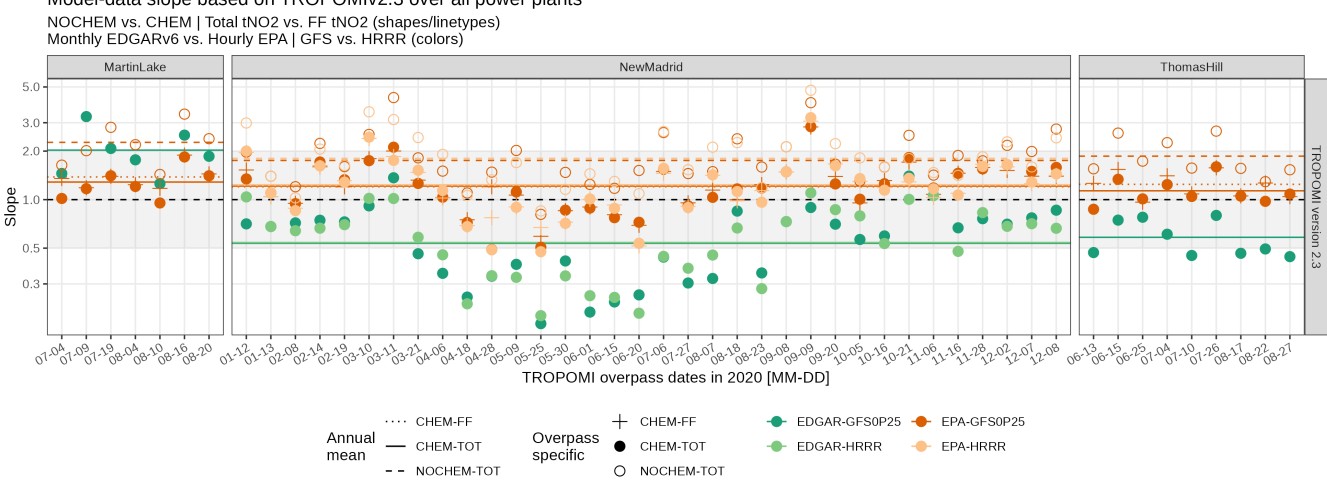

**Figure 6.** A summary figure of the linear slope between the observed $tNO_2$ and simulated $tNO_2$ using a variety of model configurations over all three US power plants. Model configurations include simulations (1) with or without $NO_x$ chemistry parameterization (empty vs. solid dots), (2) using default EDGAR or scaled emissions from EPA (green vs. orange dots), (3) using 0.25° GFS or 3km HRRR (dark green/orange vs. light green/orange dots), (4) using total $tNO_2$ or background-removed local enhancements (e.g., CHEM-FF as crosses). Annual mean slopes are displayed as horizontal solid or dashed lines. The model evaluation here uses TROPOMI v2.3 and emissions from EDGARv6.1. Evaluations based on TROPOMI v1.3 and annual mean EDGARv5 emissions are shown in **Supplement Fig. S13a**.

Statistics discussed above compared the *total* tropospheric $NO_2$ columns from the model and TROPOMI for soundings around each power plant. It is noticeable that modeled $tNO_2$ uncontaminated by emissions (i.e., background $tNO_2$) are sometimes slightly lower than observed background $tNO_2$ (**Supplement Figs. S9a, S10a**), possibly because higher chemical uncertainties are related to low-$NO_x$ regimes and non-anthropogenic $NO_x$ sources from soil and lightning are excluded from current simulations but can play a bigger role of $tNO_2$ over rural regions (Goldberg et al., 2022; Shah et al., 2022). In particular, column contributions from lightning $NO_x$ emissions aloft may be amplified since TROPOMI $NO_2$ retrieval has a higher sensitivity towards the free troposphere than PBL. Since our current model setup only accounts for anthropogenic $NO_x$ sources below 2 km, we conducted an additional test by subtracting the background $tNO_2$ from total $tNO_2$ to arrive at observed anthropogenic enhancements (the second paragraph in **Sect. 2**), assuming soundings within or outside the plumes have equal contributions from the nearby non-anthropogenic $NO_x$ sources. After subtracting the $tNO_2$ background, the model-data comparison based on observed $tNO_2$ enhancements does not change dramatically (e.g., orange crosses vs. orange solid dots in **Fig. 6**).

In summary, using more accurate $NO_x$ emissions with chemistry considerably improves the model-data comparison. Increasing the spatial resolution of meteorological fields has less impact on cases with relatively flat terrain. Larger model-data mismatches generally are associated with larger wind directional biases. Modeled values in $tNO_2$ may be slightly biased low





in summer months from April to June and high in winter months from Nov to Feb with minimal annual biases, assuming EPA
emissions and observed $tNO_2$ are unbiased.

## 4.2  Model application: two cities

We now move to city cases including an industrial city, Baotou in China, and the fast-growing city, Phoenix in the US. As
$CO_2$ and $NO_x$ are commonly co-emitted into the atmosphere, observed $XCO_2$ enhancements derived from OCO-3 Snapshot
Area Mapping (SAM) mode are displayed with observed $tNO_2$ (**Fig. 7**). Background $XCO_2$ is defined as the mean values over
the background region that is determined by $NO_2$ plumes (modified from the background approach in Wu et al., 2018). Both
cities possess relatively richer OCO-3 SAM observations co-located with TROPOMI data. Since no "true" $NO_x$ emissions are
available for cities, EDGAR is utilized as the prior emission inventory for simulating $tNO_2$ and optimizing $NO_x$ emissions.

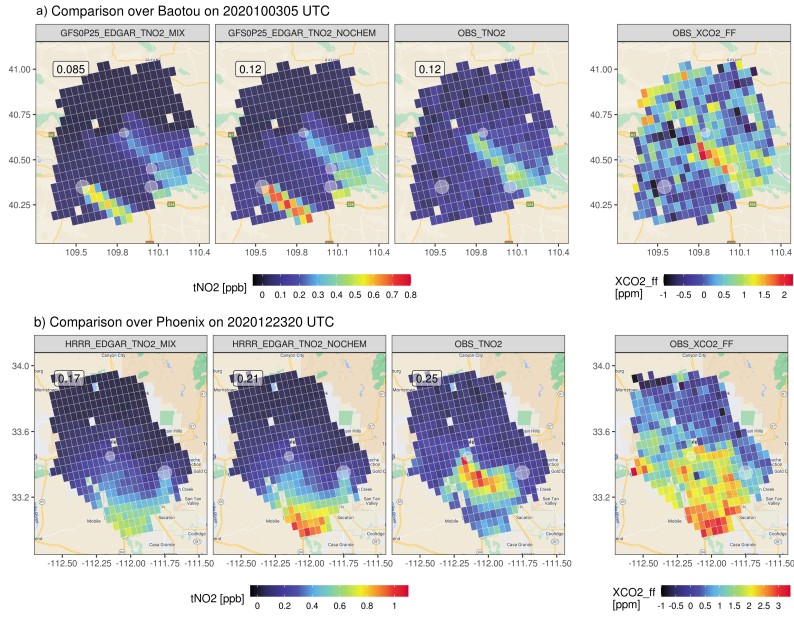

**Figure 7.** An example of GFS-based $tNO_2$ plumes over Baotou on Oct 3, 2020 (a) and the HRRR-based $tNO_2$ plumes over Phoenix on Dec 23, 2020
(b). Modeled plumes are generated using annual mean $ENO_x$ from EDGAR with top emitters highlighted in light-grey circles. Both observed $tNO_2$ and the
anthropogenic $XCO_2$ enhancements from OCO-3 are plotted. $XCO_2$ enhancements calculated from a local background have been averaged based on the
TROPOMI sounding size. Overpass time differences between TROPOMI and OCO-3 for the two cases are < 1 hour. TROPOMI observations are cropped to
match the boundary of the available OCO-3 soundings.

We simulated 18 and 12 TROPOMI overpasses respectively for Baotou and Phoenix (**Supplement Figs. S14, S15**) and
first presented one example per city in **Fig. 7**. Baotou is surrounded by four point sources suggested by EDGARv6 but one
large source in the city center informed by both the observed $tNO_2$ and $XCO_2$ enhancements on Oct 3, 2020 (**Fig. 7a**).
Such a mismatch is confirmed by the comparison of normalized $tNO_2$ across all 18 TROPOMI overpasses with various wind
speeds and directions (**Supplement Fig. S14b**), suggesting that EDGAR very likely misallocated anthropogenic $NO_x$ sources.



Similarly, the largest emission source to the east of the city center of Phoenix according to EDGAR seems suspicious and may again be misplaced once simulating more overpasses (**Supplement Fig. S15b**). The observed plume is more concentrated near
the city center compared to the HRRR-derived plume that disperses farther away from the city center on Dec 23, 2020 (**Fig. 7b**). Such a spatial offset of the $tNO_2$ plumes is likely due to an overestimation in the modeled wind speed, pushing the plumes to the southern edge while diluting $tNO_2$ values over the urban core.

When more overpasses are examined, the model captures well the seasonal variation in $tNO_2$ —i.e., higher/lower values in winter/summer months (**Supplement Figs. S14a, S15a**). Other than emission biases that affect all cases, a few overpasses stand
out for Baotou with poorer agreements with TROPOMI likely owing to (1) clear biases in wind direction on May 31, Aug 9, and Dec 15, 2020, and Feb 19, 2021, and (2) a likely overestimation in STILT footprint that may trigger several effects on Sept 29, March 29, 2020, and Oct 16, 2021 (**Supplement Fig. S14a**). Although STILT can characterize sub-grid cell turbulent mixing by its stochastic nature, the quarter-degree GFS may be insufficient to resolve the complex terrain and air flows, contributing to biases in wind directions and PBL heights over mountainous locations (Lin et al., 2017) such as over Baotou. Deviations
in PBLH may cause a cascade of effects: deciding up to what height the emissions are diluted, whether such height is above or below the emission/plume height, and chemical changes along the way. Such effects may be magnified under low-mixing low-wind conditions where the model particularly struggles with the accuracy of PBLH. Without much mixing between plume and background air, the prescribed available $O_x$ level may be overestimated adjacent to intensive $NO_x$ sources. Overestimation in $NO_2$ concentration may further be amplified considering the dependence of $NO_x$ rate changes on its concentration. Hence,
concentrations of chemically reactive species under low-mixing scenarios are extremely challenging to be modeled properly with an extreme during the nighttime.

## 5 Discussions

Our ultimate goal is to explore what can be learned about the emission characteristics from anthropogenic hotspots with the joint use of space-based $NO_2$, CO, and $CO_2$ plumes. As an intermediate step, this study is informed by previous efforts in
extracting and constraining urban $CO_2$ emissions from satellites using Lagrangian framework (Wu et al., 2018; Roten et al., 2022) and extends it to the interpretation of tropospheric $NO_2$ satellite data. To diagnose $NO_x$ emissions from $NO_2$ column signals, we need to effectively account for how $NO_x$ evolves in air parcels from its initial source to locations sampled by TROPOMI, making the Lagrangian perspective an ideal candidate. Now we discuss when and how such a framework can be of most use and possible future improvements.

### 5.1 Model advantages and flexibility

At the urban extent stretches a few hundred kilometers, our framework accounts for atmospheric transport and chemical transformation in a more rigorous way than typical statistical approaches such as the EMG method, and in a more efficient way than full-chemistry models that explicitly resolve individual chemical reactions.



Another advantage of the STILT-$NO_x$ design is that each of the three main components (trajectory calculations representing air transport, chemical production or loss of the target specie, and optimization of emission) are independent and each can reuse previously saved output from the others. For example, if one wanted to test how sensitive model concentrations were to the chosen chemical scheme, the simulations of atmospheric transport can be reused via the storage of trajectory-based modeling, thereby reducing the computational cost. Our prototype demonstrates a global solution of the $NO_x$ chemical tendency parameterized by the one set of "$NO_x$ curves" in **Fig. 3**. Although simplification may be thought of as a limitation, one can easily replace those default curves with alternatives that are tailored toward a specific region or regime of interest. Such flexibility can inform us of the influence on modeled columns from $NO_x$ curves derived from different chemical mechanisms. Similarly, one can investigate the sole meteorological influence by diversifying the meteorological and mixing parameters. Moreover, because air parcels in LPDMs are not tied to a certain atmospheric tracer, we can estimate concentrations of various species along model trajectories. It allows us to constrain emissions for multiple atmospheric constituents in a consistent framework, which may shed light on tracer-tracer analyses (**Sect. 5.2**).

The Lagrangian modeling approach has its inherent benefits. Firstly, the generation and recording of trajectories can easily reveal the source regions only relevant to a specific satellite sounding and the sub-city scale variations in emission characteristics (Wu et al., 2022). In addition to storing lat/long coordinates and extrapolated meteorological quantities along every trajectory at each timestamp, STILT-$NO_x$ outputs and records $NO_x$ concentration changes due to every process including emission, net chemical changes, and inter-parcel mixing at minute scales. Those trajectory-level concentration changes are further driven by several model configurations listed in **Sect. 4**, which facilitates model debugging and comprehends modeled results. See **Sect. 5.2** for one of the applications. Secondly, the spatial resolution of concentration calculations is not bounded by the rigid boundary of model grid cells, which is particularly important for dealing with non-linear processes for chemically active species. As demonstrated in several studies (e.g., Valin et al., 2011), the grid-average concentration may undergo excessive mixing in Eulerian models, and the concentration-driven chemical tendency varies with the adopted spatial resolution. While the Lagrangian perspective solves for concentration changes at extremely high spatiotemporal resolutions, inter-parcel mixing schemes can be implemented to "smooth" the concentration gradients, whereas it may be challenging to "recover" the sub-grid cell concentration gradients in the Eulerian framework unless increasing the spatial resolution.

More broadly, the proposed simplified parameterization of the non-linear $NO_x$ tendency or $NO_x$ curves is not limited to the STILT framework and can potentially be incorporated into other Lagrangian modeling frameworks or even Eulerian frameworks with a fine spatial resolution to resolve the local variability in chemistry.

## 5.2 Implications for constraining urban $CO_2$ emission and emission ratios

The knowledge learned from analyzing $NO_2$ plumes can be transferrable to constraining bottom-up $CO_2$ emissions. Two main sources of biases influencing the urban $CO_2$ emission constraint include biases in wind direction and emission locations. It is apparent that model-data mismatches in $NO_2$ columns have shown great value in easily identifying the biases with emission locations even without deploying atmospheric inverse analyses (**Sect. 4.2**), especially for point sources in urban areas when plumes from multiple sources are less overlapping with one another. Additionally, diagnosing the $NO_x$ emissions can facil-




itate $CO_2$ emission estimates in two ways: it can reveal systematic biases in near-field wind directions (**Sect. 5.2.1**) and by quantifying the $NO_x$ to $CO_2$ emission ratios for point and/or area sources assist in sector-based attribution (**Sect. 5.2.2**).

### 5.2.1 Quantifying wind bias

In addition to leveraging limited radiosonde measurements, the obtained modeled and retrieved $NO_2$ plumes can be used to *quantify* wind biases, to improve the accuracy of top-down $CO_2$ emission constraints, whether or not employing conventional atmospheric inversions. To do so, we conducted a second wind assessment involving a plume rotation algorithm. In brief, a $NO_2$ plume from either model or retrieval is rotated clockwise ($\alpha$ from -180 to -5° with a spacing of 5°) or counter-clockwise (from 5 to 180°) around the emission source and then resampled onto the original TROPOMI pixels (**Supplement Fig. S16**). $tNO_2$ from an original and a rotated plume are multiplied to arrive at a cross-product of $tNO_2$ in $ppb^2$, analogous to the concept of "cross-correlation". The original or the rotated plume can be chosen from either model or observations and their normalized cross-product can be expressed as a function of rotating angles (colors in **Fig. 8a**). More technical details on the intermediate steps in calculating such a function are described in **Appendix C**.

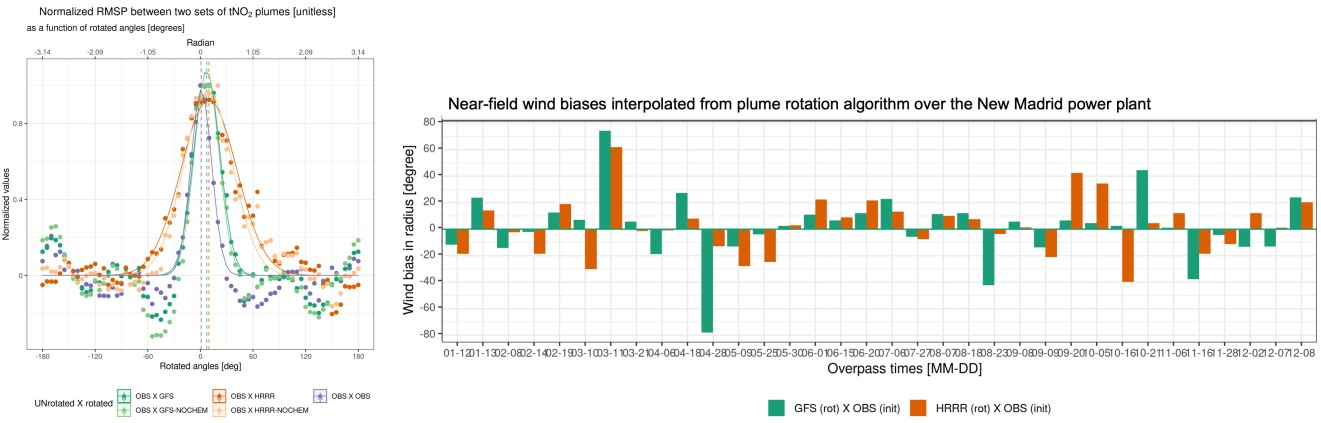

**Figure 8.** (a) An example of the normalized spatial mean of the sounding-wide product between the un-rotated observed $tNO_2$ and five different sets of rotated plumes for the New Madrid power plant on June 15, 2020. Gaussian-like curves are fitted to each set with mean and standard deviation indicating modeled wind biases. Five sets include observed $tNO_2$ (purple) and simulated $tNO_2$ driven by GFS (dark green) or HRRR (dark orange) with or without the account of $NO_x$ lifetime (light green or light orange). The horizontal dashed lines denote the $\mu$ parameter that can translate into wind bias in degrees or radians. (b) "near-field" wind directional bias quantified by the modeled $tNO_2$ plumes using 3km HRRR (orange bars) or 0.25° GFS (green bars) and retrieved $tNO_2$ plumes for every examined TROPOMI overpass (y-axis, in degrees) following the rotation algorithm in panel (a).

As a result, the width of the Gaussian-shaped curve of the cross-product (as measured by the $\sigma$ parameter of a Gaussian fit) reflects the bias in the plume shape resulting from horizontal dispersion. A larger area under the Gaussian curve indicates a greater overlap between the initial plume and the rotated plume. More importantly, deviations in the central line of the Gaussian fit away from zero (as measured by the $\mu$ parameter) imply possible biases in the "near-field" wind direction for each TROPOMI overpass (**Fig. 8b**). Specifically, wind directional biases of both GFS and HRRR appear to be smaller from May to





early Sept than the remaining months (**Fig. 8b**). A few outliers stand out due to large wind biases on March 11, April 28, Sept 20, Oct 5, Oct 16, and Nov 16, 2020.

By identifying those outliers with strong wind directional biases, one can consider either removing those cases or assigning a larger observational uncertainty when attempting to constrain emissions of $NO_x$, CO and $CO_2$, assuming their emissions are mostly co-located. Alternatively, we can use this rotating algorithm to create a model plume with minimized wind directional

bias before being fed into atmospheric inversions or data assimilation systems (which usually deal with random uncertainties). A more sophisticated approach would be to optimize the emission and wind field simultaneously (Liu et al., 2017). More investigations may be needed to examine the degree of freedom of such a wind-emission optimization framework.

### 5.2.2 Quantifying emission ratios between $NO_x$ and GHGs

Our modeling development offers additional insights into the discrepancy between emission ratios at the sources and directly-

observed enhancement ratios between two species with different chemical lifetimes. The joint use of $NO_2$ and $CO_2$ has enabled the calculation of emission ratios by adopting a spatial constant $NO_x$ lifetime (MacDonald et al., 2022; Hakkarainen et al., 2023), and the constraint of $CO_2$ emissions using $NO_2$ plumes by adopting inventory-based emission ratios (Zheng et al., 2020; Zhang et al., 2023). However, inventory-based emission ratios might not be well constrained, and the impact of how $NO_x$ decays over time and space on observed $ENO_x$–to–$ECO_2$ emission ratios have not been comprehensively assessed,

which may impair the ability to accurately quantify such observed emission ratios (Kuhlmann et al., 2021).

Thanks to the ability of our model in tracking $NO_x$ and $NO_2$ concentrations along trajectories with different model configurations, we can provide an assessment of the influence of atmospheric chemistry on estimating emission ratios. Specifically, the impact from $NO_x$ net losses from each satellite sounding (s) is specified as the ratio of modeled tropospheric $NO_x$ with chemistry over that without chemistry: $\gamma_{ts,s} = NO_{x,CHEM,s}/NO_{x,NOCHEM,s}$. Because $NO_x$ is simply treated as a passive tracer like

$CO_2$ in the NOCHEM simulations, $\gamma_{ts}$ are naturally smaller than one. Lower $\gamma_{ts}$ corresponds to faster $NO_x$ chemical frequency and more chemical losses en route to the sounding location, suggesting that $NO_2$–to–$CO_2$ enhancement ratios derived directly from satellites need to be scaled up to render the $ENO_x$–to–$ECO_2$ emission ratios at source locations.

We calculated $\gamma_{ts}$ for every sounding and present their distribution as histograms in **Fig. 9a** or as a function of the distance from the emission source (**Fig. 9b**). $\gamma_{ts}$ ranges from 0.24 to 0.61 for three power plants and from 0.42 to 0.84 for

three cities, where lower values correspond to summer months (green bars in **Fig. 9a**). That is to say, the directly observed $NO_2$–to–$CO_2$ enhancement ratios may have to be scaled up by 1.2 to even 4 times across seasons to properly "recover" the $NO_x$ being lost en route from emission sources to the sounding locations. Not properly accounting for such an effect leads to an underestimation of derived emission ratios from satellites. More importantly, discrepancies between enhancement ratios and emission ratios, reflected by $\gamma_{ts}$, are not spatially uniform. $\gamma_{ts}$ gradually decline as soundings move away from the emission

sources (**Fig. 9b**). Soundings located farther downwind from emission sources tend to undergo more chemical transformations, likely because $NO_x$ losses become more rapid as $NO_x$ concentrations become lower by atmospheric dispersion (triggering positive feedback). We clarify that only downwind soundings affected by major $NO_x$ emissions are included in **Fig. 9b**; and simulations with or without chemistry have included the effect of atmospheric dispersion as distance increases. Furthermore,



how quickly $\gamma_{ts}$ decline with distance depends on the wind speed and heterogeneity in emissions. For example, the faster the
wind may be or the more isolated emissions there are, the steeper $\gamma_{ts}$ decline with distance. $\gamma_{ts}$ at the distance of zero are
much lower than one in summer, which suggests that chemical transformation can affect the $NO_x$ inflow. We further observe
slight differences in the distribution of $\gamma_{ts}$ for cities versus power plants. Histograms of both tropospheric $NO_x$ (**Supplement
Fig. S17**) and $\gamma_{ts}$ over cities are associated with a wider spread than power plants because cities contain a wider spectrum of
emission types and intensities.

Lastly, enhancement ratios need to be adjusted considering the $NO_2$–to–$NO_x$ ratio and differences in averaging kernels
among two retrievals. The medians of our estimated $NO_x$–to–$NO_2$ ratio over power plants and cities range from 1.33 to 1.66,
which generally aligns with previous studies of around 1.32 (Beirle et al., 2011; Goldberg et al., 2022). Our estimates are lower
in winter than in summer and can be as large as 2 or 3 for a few soundings experiencing intense $NO_x$ sources (**Fig. 9c**).

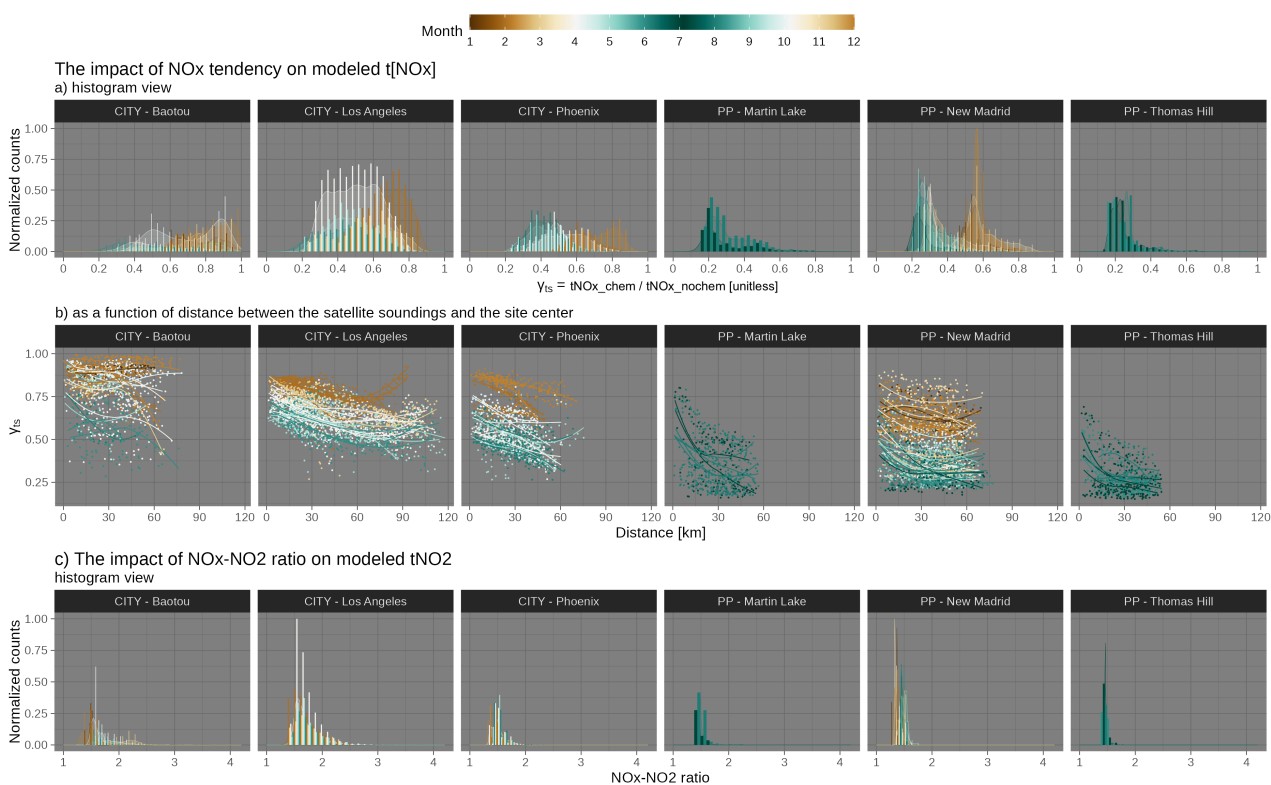

**Figure 9.** A quantitative metric of the impact from $NO_x$ chemistry on tropospheric $NO_x$ and $NO_2$ columns. Ratios in $tNO_x$ between simulations with and
without chemistry are calculated as $\gamma_{ts,s} = tNO_{x,CHEM,s}/tNO_{x,NOCHEM,s}$, which is displayed as a histogram (a) and as a function of the distance of
the satellite sounding from the site center (b). Soundings in summertime overpasses are colored in dark green whereas brown for soundings in the dormant
months. Soundings of all overpasses for all city and power plant cases are included in histograms. Only downwind soundings in the $NO_2$ plumes are included
in the distance panel (b) with a smooth spline fitted per overpass to reveal the anti-correlation. The ratio between modeled tropospheric $NO_x$ column versus
the tropospheric $NO_2$ column is derived from each sounding to reveal the $NO_2$–to–$NO_x$ ratio influence (c). As a reference, most previous studies adopted a
constant $NO_x$–to–$NO_2$ ratio (reciprocal of $NO_2$–to–$NO_x$ ratio) of 1.32 and can reach 2 in a hyper-near-field area of a major $NO_x$ source.




### 5.3 Limitation and Room for improvements

The diversity of VOCs emissions, the vertical profiles of emissions, and the extent of inter-parcel mixing may impact the modeled results. Perhaps one of the biggest limitations of the current $NO_x$ chemical representation lies in not directly accounting for VOCs, which may affect (a) the "sweet spot" on $NO_x$ curves where two $NO_x$ loss pathways reach their maximum and (b) the $O_x$ -based $NO_2$–to–$NO_x$ ratios (**Sect. 5.3.1**). Moreover, the influence of representations of emission profile on modeled $tNO_2$ can be magnified when further considering the TROPOMI $NO_2$ averaging kernel (**Sect. 5.3.2**). Simulations of point

sources like power plants may be more sensitive to these factors compared to simulations of areal sources.

### 5.3.1 The impact from $VOC_R$

To investigate the impact of VOCs on $NO_x$ curves, we calculated the VOC reactivity against OH from existing WRF-Chem results based on the following formula: $VOC_R = \sum_{i=1}^{n} k_{OH+VOC_i}[VOC_i]$ and generated separate sets of $NO_x$ curves for 4 respective $VOC_R$ intervals of [0.1, 1), [1, 3), [3, 10), and [10, 50) $s^{-1}$ with a coarse SZA bin spacing of $10°$. Curves become

much noisier at night and in pristine environments with extremely low $NO_x$ ($\leq 0.1$ ppb) where WRF-Chem /RADM2 may be less suitable (thereby not shown in **Fig. 10**). Focusing on smaller SZAs (consistent with 1 pm TROPOMI observations), the general non-linear shape of these $NO_x$ curves holds as $VOC_R$ increases (**Fig. 10**). Higher $VOC_R$ relative to the lower $NO_x$ concentration favors the oxidation of VOCs and the minor loss pathway of $NO + RO_2$ to form alkyl nitrates with a small branching ratio over the loss pathway of $NO_2 + OH$ (**Fig. 1**). To compete with the VOC oxidation in reacting with OH, the

$NO_x$ loss tendency becomes slower (**Supplement Fig. S18a**) and $NO_x$ concentrations at the optimal point (where two loss pathways reach their maximums) must be increased, as illustrated by the increasing $NO_x$ from 1.4 to 3.7 ppb in **Fig. 10**. For the same reason, the net loss timescale ($ts_{NO_x}$) generally rises as $VOC_R$ increases, e.g., from about 2 to 4 hours. To put it in context, the $NO_x$ curves shown in **Fig. 3** represent the typically curves under moderate $VOC_R$ (e.g., $< 10$ $s^{-1}$). Elevations in $ts_{NO_x}$ with $VOC_R$ may be problematic when NO is high because the $ts_{NO_x}$ has already been quite high; while more erroneous

under conditions with moderate NO and extremely high $VOC_R$ of over 10 $s^{-1}$.

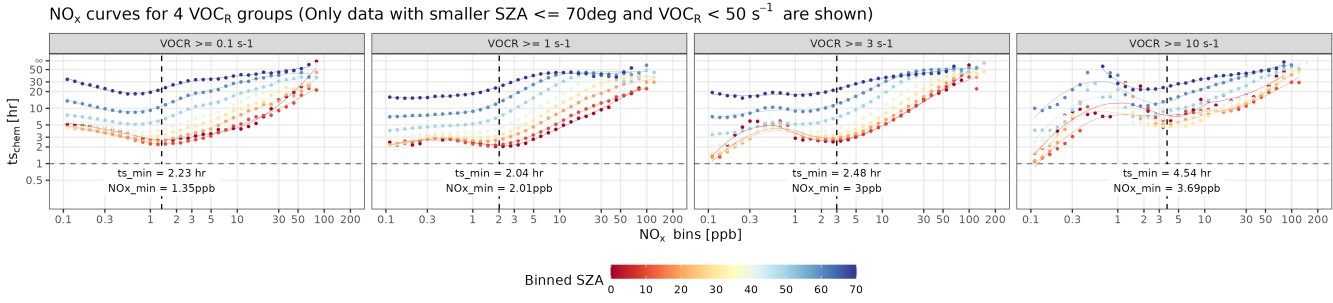

**Figure 10.** (a) Similar to **Fig. 3b**, but differentiated by 4 intervals of $VOC_R$ and SZA bins smaller than 70 degrees with an spacing of 10 degrees. All panels here utilized model results from the same WRF-Chem simulations described in Sect. 2 and **Appendix A**.



$VOC_R$ may also affect the $O_x$ level, thus, the prescribed $O_x$ of 50 ppb neglected the $O_x$ variability given its similar non-linear dependence on $NO_x$ concentrations and $VOC_R$ (Murphy et al., 2007; Li et al., 2022). Specifically, local-scale $O_x$ is primarily produced by $NO + RO_2$ or $NO + HO_2$ under $NO_x$-limited $RO_2$-rich conditions. Consequently, higher $NO_x$ concentrations result in higher $O_x$ levels. However, under $NO_x$-saturated conditions, the production of $O_x$ is suppressed by $NO_2$, leading to a decline in $O_x$ level as $NO_x$ concentration increases. For example, $O_x$ levels may be lower than 50 ppb at $NO_x$-saturated regimes. If a simplified $O_x$-$NO_x$ nonlinearity is implemented, the conversion from NO to $NO_2$ would be further slowed to alleviate strong overestimation of tropospheric $NO_2$ under stagnant mixing conditions. Nonetheless, our prescribed $O_x$ level of 50 ppb serves as a first-order cap to prevent the endless non-physical conversion to $NO_2$ from NO when $NO_x$ is extremely high since $O_3$ can be titrated.

To address these limitations, one potential approach is to leverage formaldehyde concentrations retrieved from TROPOMI. Recent studies revisited the use of the formaldehyde-to-$NO_2$ ratios (i.e., FNR) from satellites as a means of inferring $O_3$ production rates (Goldberg et al., 2022; Souri et al., 2022). Our WRF-Chem simulations, which were used to parameterize the $NO_x$ chemical tendency, show that modeled formaldehyde generally increases with $VOC_R$ with varying slopes influenced by SZA and $NO_x$ concentrations (**Supplement Fig. S18c**); and $O_3$ concentrations scale non-linearly with FNRs with $O_3$ concentration approaching a background value at high FNR > 10 (**Supplement Fig. S18d**). Even though satellite-based FNRs may theoretically help probe $O_3$ or $O_x$ concentration to better parameterize $NO_2$–to–$NO_x$ ratios, Souri et al. (2022) stressed that retrieval errors especially from formaldehyde (40 to 90% with ≤ 50% over cities) and inherent chemical errors of the predictive power of FNRs may hinder the broad application of space-based FNRs at the current stage. Nonetheless, sensitivity analyses in **Sect. 3** indicate an overall chemical uncertainty in $tNO_2$ of about 10 to 20% with respect to $NO_2$ signals, even if perturbed $O_x$ level is much lower than 50 ppb (**Fig. 4**).

### 5.3.2 The impact from emission profiles and inter-parcel mixing scales

The underlying STILTv2 (Fasoli et al., 2018) accounted for a gradual growth of the mixed layer height over the hyper-near-field area around emissions. Convolving the STILT footprint with $NO_x$ emissions, we assumed that emissions originating from the surface are uniformly mixed over the mixed layer without considering the possible uneven distribution of emissions from different vertical levels. In reality, under stable atmospheric conditions, the stack heights or plume heights of emission sources can sometimes extend above the shallow PBL. Our current assumption may thus lead to an overestimation in modeled concentrations, and such biases can in turn affect the estimate of $NO_x$ tendency. More importantly, changes in the vertical emission profile can lead to changes in concentration per model level, which also affect the tropospheric column results as the typical averaging kernel profile is far from uniform within the PBL. Recall that TROPOMI $NO_2$ AKs decreases rapidly towards the surface (**Fig. 1**). Hence, placing a plume at the surface or at an elevated altitude (e.g., 400 m) can cause a discrepancy in the modeled column values. Noticeably, Maier et al. (2022) implemented a time-varying sector-specific emission profile into STILT. Yet, the influence on column concentrations due to changes in emission profiles may require more in-depth investigations, particularly over point sources.



Accounting for inter-parcel mixing was an important aspect when developing Lagrangian chemical models. Omitting inter-
parcel mixing makes solving for non-linear processes (such as chemical $NO_x$ loss) problematic. On the contrary, Eulerian
models suffer from excessive numerical diffusion. Mixing that is too strong smooths the spatial gradient of concentration and
can lower the concentration within the fixed model grids, which may cause slight shifts in $NO_x$ regimes. Valin et al. (2011)
suggested that a spatial resolution of 4 to 12 km is sufficient to capture the non-linearity in $NO_x$ loss rate. As for Lagrangian
models, efforts can be made to enable the flux exchange between air parcels via deformations (Konopka et al., 2019; McKenna
et al., 2002). STILT realized the vertical mixing by diluting surface emissions over the ML height (Lin et al., 2003). We further
enabled an exchange in pollutants' concentrations with prescribed mixing length- and time- scales following (Wen et al., 2012).
As a final sensitivity test, we simulated $tNO_2$ using a spectrum of mixing hyperparameters over the New Madrid power plant
and found minimal influence on modeled values per sounding (uncertainty < 20%, **Supplement Fig. S19**). Uncertainties in
the prescribed mixing hyperparameters contribute even less to the modeled values over urban areas (i.e., < 10% for Phoenix
cases), where emissions are generally better mixed than at power plants.

## 6  Summary

In developing STILT-$NO_x$ , we aim to quantify anthropogenic $NO_x$ emission signals for power plants and cities using remote
sensors using a novel Lagrangian chemical system that preserves the non-linear relationship between $NO_x$ concentrations and
emissions. This development is motivated by the desire to reduce computational costs by replacing the conventional kinetics-
based approach to solve for concentration changes with a simplified parameterization relying on as few variables as possible.
Such a simplified parameterization can be improved and adopted by other Lagrangian models. This work expands the capability
of (X-)STILT in tracing the origins of chemically reactive species to simulate their concentrations at satellite soundings (**Fig.
1**). Although uncertainties exist in modeling atmospheric transport, mixing, and chemical processes, this study covers the key
$NO_x$ chemical mechanisms, various error sources, model validation, and the benefit of using $NO_x$ to constrain $CO_2$ emissions
and tracer-tracer emission ratios.

To evaluate our modeling system, which consists of the HYSPLIT-STILT core and modules of column weighting, simpli-
fied chemistry, and error analyses (**Fig. 1**), we compared modeled tropospheric $NO_2$ columns using EPA-reported emissions
against observed columns from TROPOMI over three power plants in the US (**Fig. 6**). The largest model-data discrepancies
are found for overpasses with substantial wind directional biases. Across three power plants and seasons, the systematic bias
informed by the model-data regression slope appears to be small when using EPA emissions. Switching $NO_x$ emissions from
prior emissions to EPA (usually with a scaling factor of 2 to 3) greatly improves the model-data agreement, followed by the
impact of whether to turn on the $NO_x$ chemistry. Upgrading to a higher-resolution meteorological field minimally alleviated
model-data mismatches but should be considered for regions with complex terrain. Subtracting the background $NO_2$ is nec-
essary, especially over stagnant days and regions with strong non-anthropogenic emission influences. Based on our limited
case studies, $NO_2$ simulations of power plants are usually more challenging compared to urban areas with more of the areal



source for several reasons: from atmospheric mixing, spatial heterogeneity and vertical profiles of emissions, to the exposure of ambient ozone-rich air when estimating the $NO_2$–to–$NO_x$ ratios.

Our comprehensive analyses on modeling $tNO_2$ further shed light on the estimation of $CO_2$ emissions at the local scale. Modeling two species in a consistent modeling framework makes the quantification of two key bias terms easier, namely

from wind directions and emission locations. For example, we demonstrate the use of model-data $NO_2$ plumes to obtain a quantitative value of the directional biases associated with the modeled wind (**Sect. 5.2.1**) and biases with the emission distribution in prior inventories like EDGARv6 (**Sect. 4.2**). As growing interest arise from the joint use of GHG and air pollutants, we also investigated the differences between $NO_2$–to–$CO_2$ enhancement ratios and the $ENO_x$–to–$ECO_2$ emission ratios (**Fig. 9**). Such differences between the two tracer-to-tracer ratios vary across seasons and space, which is again driven by

the non-linearity between the emissions and concentrations. For instance, to be consistent with emission ratios at the sources, observation-based enhancement ratios need to be scaled up by 2 to 3 times in the summer months due to faster photochemistry. Soundings with a separation of 60 km from the site center need to be scaled up further by roughly 1.3 times than near-field soundings concerning changes in chemical tendency.

STILT-$NO_x$ in conjunction with the forthcoming local-scale multi-tracer non-linear modeling/inversion system (**Fig. 1**) can

be employed to simultaneously constrain emissions from multiple species of both GHGs and key air pollutants along with their respective emission ratios, allowing for improvements in sectoral attributions. Such a framework has the potential to be scaled up to a large number of cities for estimating emissions of $NO_x$, $CO_2$, and possibly other tracers from space-based sensors.

*Code and data availability.* TROPOMI Level 2 $NO_2$ data and OCO-3 Level 2 B10p4r $XCO_2$ data were accessed from 10.5270/S5P-9bnp8q8 and 10.5067/970BCC4DHH24, respectively. EDGARv6.1 emissions are accessed from https://data.jrc.ec.europa.eu/dataset/

df521e05-6a3b-461c-965a-b703fb62313e. X-STILT code in modeling $NO_2$ will be archived on the GitHub repository at https://github.com/uataq/X-STILT upon publication.

## Appendix A: WRF-Chem setups

We used meteorological fields from the Global Forecast System ($0.25° \times 0.25°$ GFS-FNL, NCEP, 2015) to generate hourly outputs at a grid spacing of 12 km for five consecutive days in each month of 2020. The first day is regarded as the spin-up time

to stabilize the model whose concentration fields are excluded from the following analyses. Anthropogenic emissions of air pollutants and VOCs are adopted respectively from EDGARv6.1 (Crippa et al., 2022) and EDGARv4.3.2 (Huang et al., 2017) with biogenic VOC emissions derived from the Model of Emissions of Gases and Aerosols from Nature (MEGANv2, Guenther et al., 2012). No lightning or soil $NO_x$ source is included in WRF-Chem simulations. The boundary condition of chemicals relies on the CAM-CHEM model (Buchholz et al., 2019). The most important part is the gas phase photochemistry scheme,

which is driven by 2nd generation of the Regional Acid Deposition Model (RADM2, Stockwell et al., 1990) with Dry and wet



depositions included. RADM2 is well-suited under polluted environments but may miss several key aromatic components for pristine environments dominated by BVOCs (Stockwell et al., 1997).

**Appendix B:  Technical notes on regional wind assessment**

We assess the wind uncertainty associated with two meteorological fields that drive (X-)STILT, namely the 3 km HRRR
and 0.25° GFS. The first approach targets regional wind error statistics by comparing modeled wind fields (both HRRR and GFS) against true wind observations at radiosonde stations. The wind error statistic is further used to translate wind errors to uncertainties in $tNO_2$. Radiosonde balloons are normally launched at 00 or 12 UTC. U- and V-component wind observations for only levels below 2 km over the 24 hours ahead of the TROPOMI overpass time are selected. We then estimate random uncertainties of u-/v-component wind speed (i.e., RMSE in $m\ s^{-1}$) and normalize RMSEs over mean wind speed to yield
fractional uncertainties (%) for every overpass (**Supplement Fig. S3**). For example, fractional wind uncertainties over Missouri (around 20 to 40%) are generally smaller than uncertainties over mountainous lands in Utah (> 40%), which relates to the model's capabilities in resolving topography and topographic flows. In addition, HRRR-based winds at radiosonde stations appear to be more erroneous compared to GFS-based winds High-resolution models provide better descriptions of the surface land cover type, terrain height, and surface roughness, which may improve the spatial variability of PBLH (Lin et al., 2017) and
wind vectors. Without true wind measurements, it remains unclear whether higher-resolution models can capture more accurate fine-scale meteorology. Nevertheless, the radiosonde analysis provides an overall picture of the regional wind uncertainty.

To propagate wind error statistics to transport uncertainty in concentrations, a wind error component is added to the mean wind component and the turbulence component when generating backward trajectories. Transport uncertainties in $tNO_2$ are defined as the differences in variations of parcel-specific $NO_2$ mixing ratio with the proper vertical weighting of AK and PWF
between the perturbed and the initial set of the trajectories.

**Appendix C:  Technical notes on near-field wind bias quantification**

As introduced in **Sect. 5.2.1**, a $NO_2$ plume from either model or retrieval is rotated clockwise ($\alpha$ from -180 to -5° with a spacing of 5°) or counter-clockwise (from 5 to 180°) around the emission source and then resampled onto the original TROPOMI pixels (**Supplement Fig. S16**). We then multiply gridded $tNO_2$ from the initial plume with gridded $tNO_2$ from each rotated plume
under each rotating angle, $\alpha$. The cross-product of two $tNO_2$ plumes [XP, $ppb^2$] measures how two $tNO_2$ plumes are similar in terms of their spatial structures, as one is rotated around its source location for 360° (**Eq. C1**):

$$XP(x_s, y_s, \alpha) = tNO_2^{initial}(x_s, y_s)\ tNO_2^{rotated}(x_s, y_s, \alpha) \qquad (C1)$$

We next calculate the square-root-mean of these sounding-specific cross-products per rotating angle (**Fig. 8a**). The peak of the Gaussian shape suggests when two plume signals reach the maximum correlation, while the wing suggests when one plume signal starts to decouple with another plume signal. The plume that undergoes rotation ($tNO_2^{rotated}$) can either be a modeled
plume with different model configurations (e.g., GFS or HRRR; with or without chemistry) or an observed plume. For example,



root-mean-products (RMP) based on the simulations without chemistry displays a high bias compared to RMP using observed $tNO_2$ , which implies that the entire modeled scene including the background signal is biased high when $NO_x$ lifetime is not included. When an observed plume was rotated to match its original self, their $tNO_2$ product can serve as a baseline. Normalizing the cross-products offers a diagnostic (**Fig. 8**), analogous to the concept of "cross-correlation coefficient".

*Author contributions.* DW, POW, and JLiu designed the modeling experiments and contributed to the interpretation of results. DW realized the STILT-$NO_x$ model code and conducted WRF-Chem simulations. Specific insights from individuals: POW, JLLaughner, and PIP —$NO_x$ chemical parameterization; JLLaughner —WRF-Chem model setup; JCLin —Lagrangian inter-parcel mixing. All authors contribute to the manuscript writing and editing.

*Competing interests.* The authors declare no conflict of interest.

*Acknowledgements.* The analysis is supported by the National Aeronautics and Space Administration with grant number of 80NSSC21K1064. The computations presented here were conducted in the Resnick High-Performance Computing Center, a facility supported by the Resnick Sustainability Institute at the California Institute of Technology. We acknowledge the use of the WRF-Chem preprocessor tool of mozbc provided by the Atmospheric Chemistry Observations and Modeling Lab (ACOM) of NCAR. The authors acknowledge the NOAA Air Resources Laboratory (ARL) for the provision of the GFS and HRRR meteorological files used in this publication, which were downloaded

from the READY website (http://www.ready.noaa.gov, last access: 1 May 2018). The first author extends appreciation toward Kazuyuki Miyazaki (JPL) for discussions on $NO_x$ modeling and Rob Nelson and Annmarie Eldering (JPL) for the OCO-3 data.



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
