# Peer review of "A simplified non-linear chemistry-transport model for analyzing $NO_2$ column observations: $STILT\text{-}NO_x$"

_EGUsphere, 2023_

## Author Comment (AC2)

**Point-to-point response**

We thank two reviewers for their constructive comments and have addressed all comments with additional analyses and clarifications to the manuscript. We start the response with a summary of changes for both reviewers. Our point-to-point responses are highlighted in blue with track changes in blue/red and reviewers' comments in black.

**General responses to both reviewers**

[Figure]

Two main criticisms from the two reviewers include:

1) Omission of chemical pathways and processes (heterogeneous NOx chemistry, NO + HO2/RO2);
2) Lagrangian atmospheric mixing (mixing scales in PBL, mixing between FT and ML).

We also identified a minor bug in the estimated NOx net timescales and reran simulations. The main difference from the initial submission (**upper right panel**) is the timescale over NOx-limited regimes at high SZA bins (NOx < 0.1 ppb and SZA > 50 degrees in the **lower right panel**).

As changes in the TROPOMI version and model parameters may affect model-data comparisons, we summarize the model-data slopes for all overpasses over the New Madrid power plant based on each configuration (**Fig. R1 below**). All simulations presented here used EPA emissions and 3km HRRR meteorological fields. Similar to **Fig. 6**, not accounting for NOx chemistry results in a positive model bias of tNO2 (**blue crosses in Fig. R1**); and using the latest TROPOMI version (v2.3 from v1.3) has largely reduced the model-data mismatches (**orange circles to orange dots**). The minor correction in the derived net loss timescale and the mixing time scale (from 3 to 1hr) only slightly alter the model-data slopes (**orange dots vs. purple dots vs. purple crosses**).

[Figure]

**Figure R1.** Model-data comparisons informed by the linear slope between the two using different versions (v1/v2) are explained as follows: TROPOMI: v1.3 or v2.3 of the TROPOMI L2 NO$_2$ retrieval. CHEM: v1 vs. v2 for runs using NOx curves from the initial vs. revised manuscript (as shown in the above comparison). MIX: inter-parcel mixing with two different horizontal mixing timescales tested (3 vs. 1 hr) over a 1km box. Tests with a spectrum of mixing parameters have been conducted in response to the 1$^{st}$ comment of reviewer 2.

Comments from reviewer #1
Satellite retrievals of NO2 columns are used to determine NOx emissions from power plants and cities. They are increasingly used alongside CO2 retrievals to calculate emissions from these sources. However, the effect of NOx chemistry and transport on $NO_2$ columns is often overlooked. To address this issue, Wu et al. have developed a model that incorporates a simplified representation of NOx chemical loss within the STILT Lagrangian particle dispersion model. It includes additional features such as a column weighting module to account for retrieval averaging kernel profiles and an error analysis module. The model is evaluated against TROPOMI NO2 observations from three power plants and two cities. The manuscript covers the model's advantages, limitations, and applications such as using NO2-to-CO2 enhancement ratios to estimate CO2 emissions and identifying wind biases in meteorological data.

While this work is generally sound and well-presented, there are areas that I think need attention.
We appreciate the constructive comments from Reviewer #1 and tried to address all concerns via additional supporting analyses.

NOx chemical tendency (Sect. 2.1):
1. It appears that the model excludes heterogeneous NOx chemistry in aerosols. If this is indeed the case, it is important to discuss the resulting errors arising from this omission. Alternatively, if heterogeneous NOx chemistry is included, please clarify, and modify Fig. 1 accordingly to reflect this information.

The gas-aerosol chemistry of NOx was not turned on in WRF-Chem as properly addressing such reactions requires knowledge about aerosol loading and composition as well as uncertainties in the dependence of reaction probability on water vapor and temperature (e.g., Real et al., 2008), which remain challenging topics. Certainly, the N2O5 hydrolysis is an important NOx pathway

during nighttime. Essentially, we did not enable the arrow from N2O5 to HNO3 given our choice of WRF-Chem (see simplified diagram above). Note that RACM considers the thermal decomposition of N2O5 (Stockwell et al., 1997).

As a result, more NOx will survive the night and appear in the morning due to the photolysis and thermal decomposition of N2O5 instead of being partially lost to HNO3. The omission of N2O5 hydrolysis generally causes a high bias in $NO_2$ concentration and a resulting slow bias in the chemical loss of NOx over urban environments. In other words, the $NO_x$ chemical tendency (RNOx = PNOx- LNOx) is larger in the atmosphere (i.e., faster NOx loss rate) than in the model.

To address this comment, we added a simple sensitivity experiment to understand the impact of the omission of N2O5 hydrolysis via simple sensitivity analysis. For this experiment, we simply

assume that all existing N2O5 in the current setup of WRF-chem photolyzed into $NO_2$ during the day (SZA < 90), despite those N2O5 being extracted from the non-hydrolysis runs. We then update the bin-averaged NOx chemical tendency and compare it with the original. Reduction in chemical tendency is found for each 2-deg SZA bin and a maximum reduction occurs at SZA of ~90 degrees (**blue dots in the figure to the right**).

[Figure]

Nonetheless, considering the local overpass time of TROPOMI of ~1 pm, we expect the impact to become progressively small as the plume disperses. In other words, it becomes more of a background issue/bias, which may not greatly affect the future emission estimates if the background is properly subtracted from both TROPOMI and the model.

2.  The NOx chemical tendency is parameterized as a function of NOx concentration and the solar zenith angle. It is unclear why these were the only two variables chosen and whether they account for most of the variation in the NOx chemical tendency. Knowing the fraction of variation explained by these variables would be useful. I expected temperature (as a proxy for seasons) and the NO2/NO ratio to be potentially important variables.

We agree that we did not explicitly show to what degree the variability in NOx chemical tendency is explained by the chosen two variables. Although adding feature variables to the NOx curves will necessarily explain more variability of NOx chemical tendencies (RNOx) when training RNOx, some feature variables are difficult to "obtain" or only have a marginal effect in predicting RNOx. Our rationale is to choose environmental variables compared to atmospheric concentrations of other tracers like ozone unless possible "proxies" are available for those tracers' abundance.

To justify our choices and inform the "fraction of variation explained by these variables" as suggested by the reviewer, we conducted sensitivity analyses. The grid-level RNOx is grouped and averaged into bins of feature variables so that the variability in initial RNOx will be damped after grouping. Here we compared the initial RNOx derived from WRF-chem and the bin-averaged RNOx and reported their Pearson correlation coefficient and RMSE (**see figure shown below, now as Fig. S2a and S2b in the revised manuscript**). Those two statistical metrics were reported for every choice of variables and four scenarios with higher vs. lower NOx levels (orange vs. blue bars) and day vs. nighttime (empty bars vs. bars with stripes).

The variability in RNOx is better preserved as more variables are considered especially when NOx is high, i.e., >= 1 ppb (**orange bars in Fig. S2a**). Although the correlation remains low for lower NOx conditions (< 1 ppb), the chemical tendencies stay low, leading to small absolute random errors (**Fig. S2b**). Nonetheless, including $NO_2$/NO ratio and VOCR seems to have marginal impacts

on preserving RNOx variability over polluted regions. Introducing ozone concentration would help, but the prediction of ozone is arguably more complicated than the prediction of NOx. The correlation coefficient improves when adding air temperature, but RMSE does not largely reduce.

[Figure]

**Figure S2** (ab) Correlation coefficient and RMSE between the raw RNOx directly derived from WRF-Chem and the bin-averaged RNOx based on 8 combinations of feature variables. The number in front of the feature variables on the x-axis denotes the total # of variables used when grouping the raw RNOx. The combination we used (2-NOx+SZA for NOx concentration and solar zenith angle, SZA) is highlighted in bars with black outlines. Additional variables tested include air temperature (Ta), $NO_2$-to-NOx ratio (NN), ozone (O3), and VOC reactivity (VOCR). Results are reported separately for higher or lower NOx levels (orange or blue) during the day or night (empty bars or bars with strips).

In conclusion, we chose two variables in this version: 1) NOx concentration for initiating its non-linear dependence on chemical tendency, and 2) SZA for indicating photolysis rates. We have acknowledged that these NOx curves/relationships can be improved and replaced using more variables or different chemical schemes. In particular, the addition of background ozone if properly constrained would be an important advance (which also helps constrain the NO2-to-NOx ratio). **We now added a brief explanation in Sect. 2.1:**

The above grouping procedure of $R_{NO_x}$ based on a finite number of bins of $C_{NO_x}$ and $\theta$ unavoidably reduces the variability of $R_{NO_x}$ that were directly derived from WRF-Chem. To assess the extent to which the $R_{NO_x}$ variability can be explained by the selected binning feature variables, we performed a sensitivity test to quantify the deviation of bin-averaged $R_{NO_x}$ from the initial $R_{NO_x}$. Generally, the $R_{NO_x}$ variability is better preserved over polluted regimes with higher $NO_x$ level > 1 ppb than over low-$NO_x$ regimes (**Supplement Figs. S2ab**). Choosing $C_{NO_x}$ alone better explains the $R_{NO_x}$ variability than choosing SZA or air temperature alone. Including additional variables (e.g., air temperature, $NO_2$–to–$NO_x$ ratio, and $VOC_R$) on top of our default choice of SZA and $C_{NO_x}$ marginally improves the prediction of $R_{NO_x}$ except for the inclusion of ozone. However, estimating ozone remains a challenging problem, thereby ozone is not included as a feature variable in this study.

3. The calculation of the NOx chemical tendency was based on WRF-Chem simulations for three cities: Los Angeles, Shanghai, and Madrid. The rationale behind selecting these specific cities seems arbitrary. They are unrelated to the power plants and cities that were chosen for model evaluation. It would be helpful to have an explanation for this choice.

LA, Shanghai, and Madrid represent typical megacities in North America, Asia, and Europe and are chosen given their distinct sectoral emissions of GHGs, NOx, and VOCs. The fact that the three training cities don't overlap with the targets for predictions is in turn a benefit, which avoids possible overfitting of the NOx tendency relationships with feature variables. If we trained and then tested over the same three cities, it would be less convincing in terms of the generalization of the model parameterizations. **We add a brief explanation In Sect. 2.1**:

180 Focusing primarily on polluted environments, we carried out WRF-Chem simulations  for three mid-latitude cities and  extracted model outputs from a $2° \times 2°$ region  centered around each city. Three cities, namely Los Angeles in the US, Shanghai in China, and Madrid in Spain  represent typical megacities in North America, Asia, and Europe. Their varied climatic conditions and sectoral emissions of $NO_x$, VOC, and GHGs provide a holistic view of the variability of $NO_x$ chemical tendency. While our analyses

185 extended to power plants and cities beyond these three training sites when compared to TROPOMI data (**Sect. 4**), it helps assess the broader applicability of our chemical parameterizations.

4. The assumption of the NOx chemical tendency being independent of height (Eq. 2) is not accurate considering the vertical gradients of NOx near the surface during nighttime and early mornings. It seems important to discuss any limitations arising from this.

We apologize for the misleading text and Eq.2 and clarify that NOx chemical tendency (RNOx) is not assumed to be independent of height. Since RNOx relies on NOx concentration (CNOx) and CNOx varies vertically, RNOx varies with altitude. We clarify that both RNOx and CNOx were extracted from each WRF-Chem model grid and from 12 levels within the boundary layer. Those RNOx and CNOx were further grouped up to create those NOx curves in **Figure 3**. When it came to predicting CNOx in STILT-NOx, modeled NOx concentrations vary between STILT air parcels at different vertical levels because STILT footprint takes care of the atmospheric transport, and RNOx is further prescribed as a function of NOx concentration. **We have now modified Eq. 2 and the relevant text:**

195 By leveraging WRF-Chem's chemical diagnostic capability, we derive the net chemical tendency of $NO_x$ within each hour $[R_{NO_x}, \text{ppb hr}^{-1}]$  for every model grid within the lower 12 vertical levels (x, y, z). $R_{NO_x}$ is calculated specifically from the cumulative changes in NO and $NO_2$ concentrations solely due to chemical reactions (i.e., "chem_no2" and "chem_no" in WRF-Chem registry) following **Eqs. 2**:

$$\sum_{h_0}^{h} \Delta C_{NO_x}(x,y,z) = \sum_{h_0}^{h} \Delta C_{NO}(x,y,z) + \sum_{h_0}^{h} \Delta C_{NO_2}(x,y,z) \tag{2a}$$

200
$$R_{NO_x}(x,y,z,h) = P_{NO_x}(x,y,z,h) - L_{NO_x}(x,y,z,h) = \frac{\sum_{h_0}^{h} \Delta C_{NO_x}(x,y,z) - \sum_{h_0}^{h-1} \Delta C_{NO_x}(x,y,z)}{1 \text{ hr}} \tag{2b}$$

where model hour h denotes the  start time of each hour interval in the WRF-Chem outputs and z denotes the index of model vertical levels (i.e., from 1 to 12). $\sum_{h_0}^{h} \Delta C_{NO_x}$ describes the cumulative net changes to $NO_x$ concentration given chemical reactions from the initial model hour $h_0$.

5. It would be useful to assess the consistency of the NOx chemical lifetime from WRF-Chem to the available observations, although limited.

We agree that assessing WRF-Chem against observations would be useful. However, the main goal of this study is not to determine if the RACM2 chemical scheme in WRF-Chem is correct, but to investigate if a simplified model can replace a sophisticated chemical scheme, assuming it is correct. There had been numerous studies aiming to evaluate and improve WRF-Chem chemical schemes. Lastly, we recognize that our simplified NOx curves can be improved and replaced.

NO2-to-NOx ratio (Sect. 2.2):
1. The reactions of NO + HO2 and NO + RO2 to form NO2 are excluded, but they are important in the boundary layer.

We agree that RO2/HO2 are additional sources to oxidize NO, which is missing from the estimate of the $NO_2$-to-NOx ratio (that relies on modeled NOx values and a prescribed Ox level of 50 ppb, which neglects local Ox variability with VOC reactivity as discussed in **Sect. 5.3.1**). Properly resolving such reactions requires knowledge/input of VOC emission and chemistry, which would be even more challenging and essentially requires a full chemistry model. Yet, there may be room for simple parameterizations as further discussed in **Sect. 5.3.1,** including 1) adding background ozone, tracking ozone or its production rate along trajectories by leveraging satellite column observations of HCHO, and 2) parameterizing the $NO_2$-to-NOx ratio as a function of feature variables including VOCR. **Relevant modifications to Sect. 5.3.1 are attached as follows:**

$VOC_R$ may also affect the $O_x$ level  and the $NO_2$–to–$NO_x$ ratio. The prescribed $O_x$ level of 50 ppb  (**Sect. 2.2**) overlooks the nonlinear $O_x$ variability  related to $VOC_R$ (Murphy et al., 2007; Li et al., 2022).  In $NO_x$-limited scenarios, OH favors the oxidation of VOCs, and

630 local-scale $O_x$ is  predominately produced by NO + $RO_2$ or , suggesting higher $O_x$ levels with increased $NO_x$ concentrations The omission of NO + $RO_2$ or $HO_2$ in **Eqs. 3** could lead to an underestimation of the $NO_2$–to–$NO_x$ ratio, which likely explains the modeled $tNO_2$ being consistently lower than observations over background regions. Conversely, under $NO_x$-saturated conditions, the  consumption of OH by $NO_x$ may limit the VOC oxidation and $O_x$ production, leading to

635 a decline in $O_x$ level as $NO_x$ concentration  rises. Consequently, the $NO_2$–to–$NO_x$ ratio might be overestimated when true $O_x$ levels  fall below 50 ppb  (particularly under stagnant atmospheric mixing) or underestimated due to the absence of NO + $RO_2$ reactions. Nevertheless, our pre-determined $O_x$  would be further slowed to alleviate strong overestimation of tropospheric under stagnant mixing conditions. Nonetheless, our prescribedservescap~~

640  limit to prevent unrealistic conversion from NO  from NO when NO_2 at extremely high $NO_x$  levels when $O_3$  is being titrated.

To address these limitations, one potential approach is to leverage formaldehyde concentrations retrieved from TROPOMI. Recent studies revisited the use of the formaldehyde-to-$NO_2$ ratios (i.e., FNR) from satellites as a means of inferring $O_3$ production rates (Goldberg et al., 2022; Souri et al., 2022). Our WRF-Chem simulations, which were used to parameterize the

645 $NO_x$ chemical tendency, show that modeled formaldehyde generally increases with $VOC_R$ with varying slopes influenced by SZA and $NO_x$ concentrations (**Supplement Fig. S18c**); and $O_3$ concentrations scale non-linearly with FNRs with $O_3$ concentration approaching a background value at high FNR > 10 (**Supplement Fig. S18d**). Even though satellite-based FNRs may theoretically help probe $O_3$ or $O_x$ concentration to better parameterize $NO_2$–to–$NO_x$ ratios, Souri et al. (2022) stressed that retrieval errors especially from formaldehyde (40 to 90% with $\leq$ 50% over cities) and inherent chemical errors of the predictive

650 power of FNRs may hinder the broad application of space-based FNRs at the current stage. Nonetheless, sensitivity analyses in **Sect. 3** indicate an overall chemical uncertainty in $tNO_2$ of about 10 to 20% with respect to $NO_2$ signals, even if perturbed $O_x$ level is much lower than 50 ppb (**Fig. 4**).

2. Line 242: Please clarify how the NOx chemical tendency in the model change when ozone is titrated near high emitters.

We clarify that in STILT-NOx, only NOx (no ozone) is tracked and updated per timestamp along the trajectory; and the calculation of NOx chemical tendency relies on NOx concentrations and SZA. Thus, RNOx would not change as ozone decreases in the model. See our sensitive analysis above about the performance of estimating RNOx using additional variables (see our earlier response to the second comment).

The initial text was to explain how the $NO_2$-to-NOx ratio parameterized separately from RNOx would decrease when NOx emission/concentrations become extremely high (titrating ozone). Such titrations near high NOx emitters are controlled by the prescribed total oxidant level, [Ox] = [O3] + [$NO_2$], despite a simple constant value being assigned (see response above for limitation of this constant Ox level). For example, if a STILT-$NO_x$ air parcel passes by a larger emitter like a power plant, its tagged NOx concentration will be largely enhanced but its final $NO_2$ column at the receptor/sounding location may stay low given limited oxidant capacity.

Eq 1: The processes of NO2 dry deposition and mixing between the mixed layer and the free troposphere seem to be neglected.

We thank the reviewer for pointing out those physical processes. We agree that neglecting the dry deposition of $NO_2$ or NO may overestimate their concentrations and alter the estimated chemical tendency of NOx along the trajectories that further feedback to the concentrations. Previous studies implemented the deposition module that relies on the key estimation of "dry deposition velocity" for air parcels residing within the lower 50 m from the surface layer (e.g., Wen et al., 2011). Moreover, the mixing between the mixed layer and the free troposphere (FT) was neglected and requires future model modifications, such as 1) quantification of the entrainment zone and 2) the choice of vertical mixing time-/length- scales at the PBL-FT interface. Although issues with mixing and deposition were not fully resolved in this study, **we acknowledge these limitations and discuss possible future improvements in Sect. 5.3.2:**

**5.3.2** Uncertainties in non-chemical processes

[revised manuscript text omitted]

Eq. 5 assumes that the model transport and chemistry errors are independent, when in fact these errors are related.

While it is difficult to fully separate the transport error from the chemistry error (since the physical and chemical processes are fully coupled), our transport error analysis that involves perturbing wind fields and recalculating [NOx] along trajectories implicitly contains errors in tNO2 due to errors in wind and chemistry. This can be explained as follows:

Imagine a modeled air parcel/trajectory passing over a point source in the default run. With a small perturbation to the model winds, the air parcel no longer passes over that point source. The NOx concentration of these two air parcels would be different for two reasons:

1) the wind fields are perturbed → leading to changes in emissions and NOx concentrations (e.g., lower values for the perturbation run); and
2) the associated feedback involving chemistry → lower NOx concentration in the end simulation leads to changes in chemical tendency (and so on for further timesteps).

Our linear combination of three error sources would likely have been a conservative estimate given the nature of the transport error analysis. An alternative approach is to perturb the wind fields and the chemistry simultaneously to account for the covariance between the two sources of uncertainties, which is hard to realize with existing tools.

Fig. 5 and elsewhere: Please clarify how tNO2, the tropospheric NO2 column, is converted to a volume mixing ratio.

Thanks for this comment. We clarify that "tNO2" denotes the tropospheric $NO_2$ mixing ratio that is derived from the initial tropospheric $NO_2$ vertical column density (VCD). **We now add a brief explanation to the introduction section when tNO2 is first brought up:**

In this study, we present a non-linear modeling framework, STILT-$NO_x$, to simulate tropospheric column-average $NO_2$  mixing ratio ($tNO_2$) as retrieved from TROPOMI. Note that initial $NO_2$ vertical column density [VCD, molec $cm^{-2}$] is converted to $tNO_2$ [ppb] by dividing by a dry air VCD. The dry air VCD is calculated by integrating a profile of the ideal gas number density of air minus a modeled water vapor profile.

Line 382: Considering the model's sensitivity to errors in the wind direction and speed, it may be better to use winds from reanalyses datasets or to bias correct the model using nearby observations. This seems important for the inversion work planned in the future. Are there other ways to reduce this error?

Yes – as emphasized in Sect. 5, wind biases affect the interpretation of the model-data alignment of column $NO_2$ and other species and possibly the corresponding inverted fluxes/emissions. We note that meteorological reanalyses with relatively coarse spatiotemporal resolutions are not real observations and are associated with uncertainties. For example, the GFS meteorological field driving STILT is a data assimilation (reanalysis) product but still cannot resolve highly fine-scale wind patterns. Moreover, the nearby observations are spatially limited, e.g., the radiosonde observations used for regional wind assessment and transport error estimates (Sect. 3).

We may further differentiate random wind errors from systematic wind biases. The former error can be quantified (Sect. 3) and usually be propagated into the observational error matrix during the inversion. It is the wind biases that have hardly been addressed in inversion studies. Possible approaches as we discussed in Sect. 5.2.1 include the rotation of plumes or a more sophisticated data assimilation approach to correct emissions and wind altogether (e.g., Liu et al., 2017). In the ongoing inversion work, we will design and conduct OSSE to investigate the quantitative impact of wind biases on inverted fluxes and explore ways to explicitly address the wind error.

Lines 564-570: These lines are unclear.

As shown in Fig. 1, the two essential competing loss pathways of NOx are $NO_2$ + HO-> HNO3 and NO + RO2-> RONO2 with a minor branching ratio (typical value of 0.04 and further depends on

the R groups). In other words, when VOCR stays high and NOx is limited, there is a tendency to form RONO2 over HNO3, which explains why the trough of the NOx curves shifts to the higher end of NOx concentration as shown in Fig. 10. We have reworded these lines as follows:

When considering lower SZAs (consistent with  TROPOMI overpass time of 1 pm  local time), the

615 general non-linear  characteristic of these $NO_x$ curves holds as $VOC_R$ increases (**Fig. 10**). Higher $VOC_R$ relative to  lower $NO_x$ concentration favors the oxidation of VOCs  by OH and the associated minor loss pathway of $NO + RO_2$ to form alkyl nitrates with a  over the minor branching ratio, over the competing major $NO_x$ loss pathway of $NO_2 + OH$ (**Fig. 1**).  With rising $VOC_R$, the $NO_x$  chemical tendency becomes more positive (P - L, **Supplement Fig. S18a**) and

620  the net loss timescale elongates (e.g., ts_min from 2 to 4 hours in **Fig. 10**.  ). Moreover, $NO_x$ is required to reach a higher level to compete with the reactions involving VOCs, evident by the shift in the trough of the $NO_x$ curves (e.g.,  NOx_min from 1.4 to 3.7 ppb in **Fig. 10**). To put it in context, the $NO_x$ curves shown in **Fig. 3** represent  typical patterns as long as

625 $VOC_R$  remains below $10\ s^{-1}$

[Figure]

**Figure 10.** (a) Similar to **Fig. 3b**, but differentiated by 4 intervals of $VOC_R$ and SZA bins smaller than 70 degrees with  a spacing of 10 degrees. All panels here utilized model results from the same WRF-Chem simulations described in Sect. 2 and **Appendix A**.

---

## Author Comment (AC4)

**Point-to-point response**

We thank two reviewers for their constructive comments and have addressed all comments with additional analyses and clarifications to the manuscript. We start the response with a summary of changes for both reviewers. Our point-to-point responses are highlighted in blue with track changes in blue/red and reviewers' comments in black.

[Figure]

**General responses to both reviewers**

Two main criticisms from the two reviewers include:
1) Omission of chemical pathways and processes (heterogeneous NOx chemistry, NO + HO2/RO2);
2) Lagrangian atmospheric mixing (mixing scales in PBL, mixing between FT and ML).

We also identified a minor bug in the estimated NOx net timescales and reran simulations. The main difference from the initial submission (**upper right panel**) is the timescale over NOx-limited regimes at high SZA bins (NOx < 0.1 ppb and SZA > 50 degrees in the **lower right panel**).

As changes in the TROPOMI version and model parameters may affect model-data comparisons, we summarize the model-data slopes for all overpasses over the New Madrid power plant based on each configuration (**Fig. R1 below**). All simulations presented here used EPA emissions and 3km HRRR meteorological fields. Similar to **Fig. 6**, not accounting for NOx chemistry results in a positive model bias of tNO2 (**blue crosses in Fig. R1**); and using the latest TROPOMI version (v2.3 from v1.3) has largely reduced the model-data mismatches (**orange circles to orange dots**). The minor correction in the derived net loss timescale and the mixing time scale (from 3 to 1hr) only slightly alter the model-data slopes (**orange dots vs. purple dots vs. purple crosses**).

[Figure]

**Figure R1.** Model-data comparisons informed by the linear slope between the two using different versions (v1/v2) are explained as follows: **TROPOMI**: v1.3 or v2.3 of the TROPOMI L2 $NO_2$ retrieval. **CHEM**: v1 vs. v2 for runs using NOx curves from the initial vs. revised manuscript (as shown in the above comparison). **MIX**: inter-parcel mixing with two different horizontal mixing timescales tested (3 vs. 1 hr) over a 1km box. Tests with a spectrum of mixing parameters have been conducted in response to the 1st comment of reviewer 2.

**Comments from reviewer #2**
This paper introduces STILT-NOx, a Lagrangian chemical transport model, evaluates it against satellite-based column observations of NOx, and presents various sensitivity studies. The paper is well written, and I recommend publishing after the following minor comments are addressed.

We thank reviewer #2 for the positive feedback and have tried to address all the comments. Please also refer to the general comments for both reviewers on the first page if neccesary.

General comments:
Interparcel-mixing: I was a bit surprized to see the 3-hour timescale for mixing within a volume with a horizontal of 1km x 1km and a vertical extend from surface to PBL top described as "relatively fast mixing". Given the fact that the plume emitted from a power plant, when transported over 3 hours (or 30 km for typical winds speeds of 10 m/s within a well-developed PBL), and given the shape of typical plumes seen from satellite imagery, can the mixing time scale not be estimated from that? Based on that I would expect that within three hours mixing likely occurs over much larger "boxes". This is along the thought that dispersion/mixing is similar when running LPDMs backward and forward as shown in the Lin et al. 2003 paper, so forward mixing as needed for chemistry should be similar to backward mixing (or spreading of particles emitted at the same time). Would a smaller mixing timescale increase the impact of mixing, as with a 3-hour time the impact was found to be less than 5% (line 356)? Was that assessed when comparing no-mixing (i.e. infinite timescale) with mixing turned on? I think this deserves a bit of attention, as it keeps puzzling me that given the quite nonlinear property of NOx chemistry mixing at those scales does not seem to matter in chemistry-transport simulations.

What matters here is the horizontal turbulent mixing/diffusion scales as the rapid vertical mixing in STILT is realized by diluting emissions from the surface to the mixed layer. If horizontal mixing is not accounted for, individual air parcels will not "communicate" with their neighbor parcels for mass exchange. Indeed, previous studies have used a Gaussian plume model to fit the observed plumes, especially of CO, to determine the horizontal diffusion rate from observed concentrations and their horizontal distribution (e.g., along the across-wind direction). However, the chemical decay and column observations of $NO_2$ make it challenging to accurately separate diffusion timescales from chemical timescales.

To address the question of inter-parcel mixing, we have added a sensitivity test of modeled tropospheric $NO_2$ to various horizontal mixing length- or time- scales. Three mixing length scales of 1, 3, and 10 km are tested, considering the typical km scale of satellite sounding, emission grid, or high-resolution meteorological fields.

According to Seinfeld and Pandis (2016) and several practical horizontal diffusion treatments (e.g., either the simple constant diffusion coefficient or the Smagorinsky scheme in WRF), the typical diffusion time scale ranges from minutes to a few hours for a mixing box of 1 km. For example, Seinfeld and Pandis (2016) suggested that under stable met conditions, the horizontal diffusion coefficient $K_{yy}$ is on the order of 50 to 100 $m^2$ $s^{-1}$, which translates to approximately 2.7 hrs in terms of a characteristic diffusion timescale ($dx^2 / K_{yy}$).

Furthermore, we tested a range of time scales from 0.1 to 100 hours over three receptors from a summertime New Madrid power plant overpass (see plot shown below, now as Fig. S20). Three receptors differ in the number of trajectories influenced by the single power plant emission. Since STILT-NOx tracks modeled tNO2, xCO, and xCO$_2$ with mixing and without mixing, we can calculate the normalized difference in the modeled tNO2 between the mixing runs and the non-mixing runs, i.e., = ( tNO2_mix − tNO2_nomix ) / tNO2_nomix.

[Figure]

**Figure S20.** Normalized difference in modeled tropospheric NO$_2$ between the mixing and non-mixing runs as a function of the horizontal mixing timescale [hr, x-axis] and mixing length scale [km, in colors]. The normalized difference is calculated as $(tNO_{2,MIX} - tNO_{2,NOMIX})/tNO_{2,NOMIX}$. Three examples for three receptors/soundings on June 15th for the New Madrid case are shown here and they differ by the fraction of model trajectories that "hit" the power plant emission. For receptors where some trajectories encountered the emissions, a faster mixing reducing the spatial gradient in NO$_x$ leads to a reduced final tNO$_2$ at the receptor (left two panels).

As a result, for the receptor close to the power plant emission (where 61% of the trajectories were affected by power plant emissions for at least 1 min, in the middle panel in Fig. S20), the reduction in tNO2 is as minimal as about 2% regardless of the mixing length scale. In contrast, when trajectories are not influenced by power plant emissions (i.e., receptor/sounding sitting on the edge of the plume, in the left- or right-most panels in Fig. S20), the changes between mixing and non-mixing simulations appear to be larger, i.e., reaching 5%. The larger discrepancy is reasonable as mixing would exchange the NOx concentration between trajectories experiencing the plume and trajectories in the background.

In summary, mixing tends to "smooth" the gradient in NOx concentrations among air parcels but to a less extent if a larger fraction of the total 2000 air parcels is contaminated by the plume.

We now added relevant discussions of horizontal PBL mixing and other mixing in Sect. 5.3.2 (see track changes attached on the following page).

Accounting for inter-parcel mixing was an important aspect when developing Lagrangian chemical models. Omitting inter-parcel mixing makes solving for non-linear processes (such as chemical $NO_x$ loss) problematic. On the contrary, Eulerian models suffer from excessive numerical diffusion. Mixing that is too strong smooths the spatial gradient of concentration and

675 can lower the concentration within the fixed model grids, which may cause slight shifts in $NO_x$ regimes. Valin et al. (2011) suggested that a spatial resolution of 4 to 12 km is sufficient to capture the non-linearity in $NO_x$ loss rate. As for Lagrangian models, efforts can be made to enable the flux exchange between air parcels via deformations (Konopka et al., 2019; McKenna et al., 2002).  In addition to the inter-pacel mixing within the mixed layer (ML), several other turbulent mixing processes require future investigation, including (1) horizontal mixing in the free troposphere (FT), (2) vertical mixing

680 between the ML and FT, and (3) mixing between tracked air parcels with the untracked surrounding background. For example, Real et al. (2008) utilized a linear relaxation with exponential decay of the plume concentrations towards the background based on a timescale of 2 days to address the third mixing process. The second mixing process requires future modifications involving the determination of entrainment zones and mixing hyperparameters for such ML-FT exchange.

The original STILT model realized vertical mixing by diluting surface emissions  across the ML height (Lin et al.,

685 2003)  and we further enabled an exchange in pollutants' concentrations with prescribed mixing length- and  time-scales representing typical horizontal mixing rates (**Sect. 2.3**). As final sensitivity tests, we simulated $tNO_2$  based on a spectrum of mixing hyperparameters  for the New Madrid power plant . Uncertainties in the mixing parameters result in minimal uncertainties on the sounding-level modeled $tNO_2$ values (**Supplement Fig. S19**). For example,

690 differences in modeled $tNO_2$ between the mixing and non-mixing simulations become larger as mixing becomes faster and for receptors/soundings located on the edge of the plume (i.e., only a small fraction of the trajectories encountered power plant emission in **Supplement Fig. S20**). Uncertainties in the prescribed mixing hyperparameters contribute even less to the modeled values over urban areas (i.e., < 10% for Phoenix cases), where emissions are generally better mixed than at power plants. In addition, such mixing influence can vary with the spatial resolution of the emission inventory used in the simulations.

695 The dry deposition of $NO_2$ was not factored into this study, which could lead to an overestimation of modeled $NO_2$. For future model implementations, it is possible to track loss of $NO_2$ concentrations due to dry deposition by calculating "dry deposition velocities" (e.g., ?) when air parcels descend close to the surface, e.g., 50 meters above the surface (Wen et al., 2012).

Rotation of wind: As the wind changes significantly within the atmospheric boundary layer with height (the Ekman spiral), differences between modelled wind direction and the direction apparent from the observed plume can also be related to inaccuracies in the plume release height distribution, potentially associated with plume rise of the buoyant exhaust. I would recommend this to be discussed.

We have overlooked the impact of inaccurate emission profiles on the modeled plume shapes (not just on column concentrations) and quantified near-field wind directional biases. One can examine the vertical wind shear from trajectories output for the near-field region between receptor and emission sources for more clues.

We now add the following text in Sect. 5.3.2 (the limitation section on emission profiles and mixing).

**5.3.2** Uncertainties in non-chemical processes

Besides simplification of chemical reactions, modeled tNO$_2$ values can be subject to a few physical processes and parameters,
655   including emission profiles, inter-parcel mixing scales, and dry deposition.

The underlying STILTv2 (Fasoli et al., 2018) accounted for a gradual growth of the mixed layer height over the hyper-near-field area around emissions.  Yet, by convolving the STILT footprint with NO$_x$ emissions, we assumed that emissions  originate from the surface and are uniformly mixed over the mixed layer without considering the possible uneven distribution of emissions from different vertical levels. In reality, under stable atmospheric conditions, the stack heights
660   or plume heights of emission sources can sometimes extend above the shallow PBL. Our current assumption may thus lead to an overestimation in modeled concentrations, and such biases can in turn affect the estimate of NO$_x$ tendency. More importantly, changes in the vertical  profile of emissions can lead to changes in concentration per model level, which  affect the tropospheric  columns as the typical averaging kernel profile is far from uniform within the PBL. Recall that TROPOMI NO$_2$ AKs decreases rapidly towards the surface (**Fig. 1**). Hence, placing  an emission plume at the surface or
665    an elevated altitude (e.g., 400 m) can cause a discrepancy in  modeled column concentrations. In addition, if the wind shear is strong over an intensive point source (likely the Intermountain example in **Fig. 5c**), assumptions in the injection height and vertical profile of emission plumes may affect the modeled plume shape and possibly deviate the estimated near-field wind bias following **Sect. 5.2.1**. Noticeably, Maier et al. (2022)  investigated the influence of inaccurate representation of emission profiles on the flask-like modeled concentrations by implementing a time-varying sector-
670   specific emission profile into STILT.  Such an impact on column concentrations  may be minimized but yet requires future in-depth investigations, particularly over point sources.

Specific comments:

Fig. 3: which WRF-Chem runs were used? Before three different cities were mentioned, are all those simulations included in Fig. 3?

Yes – chemical tendency derived from every WRF-Chem model grid around the three cities (LA, Madrid, and Shanghai) from all monthly simulations (4 days in each month across Jan to Dec with the first day as the spin-up time) are included and aggregated into SZA and NOx bins as shown in Fig. 3. We did remove model grids with extreme values (timescale > 72 hours) to avoid skewing the average values per bin.

L200: Were the WRF-Chem simulations also selected for cloud-free conditions?

Good question- we did not explicitly remove cloudy scenes from WRF-Chem. During revisions, we re-examined the cloud mixing ratio estimated by WRF-Chem to address the impact of cloud covers on the non-linear RNOx ~ NOx + SZA relationship. For Shanghai or LA, the average fraction of cloudy pixels ranges from 0.03% (or 0.02%) in winter to 3.6% (or 4.8%) in the spring/summer months. On an annual basis, the cloudy pixels only occupy ~1.8% to 2.1% of the select near-field model pixels for three cities. We might expect the impact on NOx curves to be minimal especially since millions of grid cells are considered to create the NOx curves.

Fig. 5 b and d: the color code is missing, or am I overlooking something?

The color legend for Fig. 5b and 5d is labeled at the bottom of Fig. 5d. The four different colors represent linear fit and slope when TROPOMI is compared with four STILT-NOx runs using 1)

meteorological fields (GFS vs. HRRR), 2) emission (EDGAR vs. EPA), and 3) chemistry (default chemistry vs. non-chemistry).

L394: in table S1 the RMSE values range from 0.11 to 0.25 ppb

We clarify that range of RMSE is reported for only the simulations with NOx chemistry enabled. The highest RMSE of 0.25 ppb corresponds to simulations without NOx chemistry. **We modify the text as:**

> slightly high biased (regression slope up to 1.2, **Table S1**).  RMSE values between
>
> 425  observed and modeled $tNO_2$  when enabling $NO_x$ chemistry range from 0.11 to 0.15 ppb (Table S1), which is comparable to the random uncertainty in the $NO_2$ retrieval of 0.09 ppb.

L412: "fast-growing" is relative, Baotou certainly has a faster growth in population than Phoenix

Corrected – Phoenix sees a high population growth rate of 2.5% since 2020, one of the "fast-growing" cities in the US. **We now revised the text as:**

> We now move to city cases including an industrial city, Baotou in China, and  one of the fastest growing megacities in the US, Phoenix. As $CO_2$ and $NO_x$ are commonly co-emitted into the atmosphere, observed $XCO_2$ en-

Fig. 8a: What does RMSP stand for?

Corrected – the y-axis on Fig. 8a represents the spatial mean of the sounding-wise products of two $NO_2$ plumes with the former one being the initial unrotated observed plume and the latter one being the rotated plume (either from TROPOMI or STILT-NOx using GFS or HRRR). We've now fixed the title and caption for Fig 8a.

Table 1 in the supplement should be named "Table S1"
Corrected.

Fig. S6: please use axis titles that clearly indicate v1 and v2 in all figures
Corrected.

Fig. S7: the symbols in the legend don't quite fit with those in the figure. Triangles should be should only for EDGAR estimates, not for EPA.
Corrected.

Fig. S10 a and b: please use fewer x-axis labels
Corrected.